# Origin and evolution of qingke barley in Tibet

Xingquan Zeng[1,2], Yu Guo[3], Qijun Xu[1,2], Martin Mascher [4], Ganggang Guo [5], Shuaicheng Li[6], Likai Mao[3], Qingfeng Liu[3], Zhanfeng Xia[3], Juhong Zhou[3], Hongjun Yuan[1,2], Shuaishuai Tai [3], Yulin Wang[1,2], Zexiu Wei[1,7], Li Song[3], Sang Zha[1,2], Shiming Li[3], Yawei Tang[1,2], Lijun Bai[8], Zhenhua Zhuang[8], Weiming He[3], Shancen Zhao[3], Xiaodong Fang [3], Qiang Gao[3], Ye Yin[3], Jian Wang[9,10], Huanming Yang[9,10], Jing Zhang[5], Robert J. Henry [11], Nils Stein [4] & Nyima Tashi[1,7]

Tibetan barley (*Hordeum vulgare* L., qingke) is the principal cereal cultivated on the Tibetan Plateau for at least 3,500 years, but its origin and domestication remain unclear. Here, based on deep-coverage whole-genome and published exome-capture resequencing data for a total of 437 accessions, we show that contemporary qingke is derived from eastern domesticated barley and it is introduced to southern Tibet most likely via north Pakistan, India, and Nepal between 4,500 and 3,500 years ago. The low genetic diversity of qingke suggests Tibet can be excluded as a center of origin or domestication for barley. The rapid decrease in genetic diversity from eastern domesticated barley to qingke can be explained by a founder effect from 4,500 to 2,000 years ago. The haplotypes of the five key domestication genes of barley support a feral or hybridization origin for Tibetan weedy barley and reject the hypothesis of native Tibetan wild barley.

[1] State Key Laboratory of Hulless Barley and Yak Germplasm Resources and Genetic Improvement, Lhasa 850002, China. [2] Research Institute of Agriculture, Tibet Academy of Agriculture and Animal Husbandry Sciences, Lhasa Tibet 850002, China. [3] BGI Genomics, BGI-Shenzhen, Shenzhen 518083, China. [4] Leibniz Institute of Plant Genetics and Crop Plant Research (IPK) Gatersleben, 06466 Seeland, Germany. [5] Institute of Crop Science, Chinese Academy of Agriculture Sciences, Beijing 100081, China. [6] Department of Computer Science, City University of Hong Kong, Hong Kong 999077, China. [7] Tibet Academy of Agriculture and Animal Husbandry Sciences, Lhasa Tibet 850002, China. [8] Chengdu Life Baseline Technology Co., Ltd., Chengdu 610041, China. [9] BGI-Shenzhen, Shenzhen 518083, China. [10] James D. Watson Institute of Genome Sciences, Hangzhou 310058, China. [11] Queensland Alliance for Agriculture and Food Innovation, University of Queensland, Brisbane, QLD 4072, Australia. These authors contributed equally: Xingquan Zeng, Yu Guo. Correspondence and requests for materials should be addressed to N.S. (email: stein@ipk-gatersleben.de) or to N.T. (email: nima_zhaxi@sina.com)

Barley (*Hordeum vulgare* L.) is one of the founder crops of Old World agriculture and probably the first crop cultivated by humans[1]. At present, most barley is used as animal feed, malt, or as a component of various health foods[2]. Called "qingke" in Chinese or "nas" in Tibetan, six-rowed hulless (or naked) barley has been used as a major staple food of Tibetans for generations[3–5]. Some studies have suggested that in addition to a center of domestication in the Near East, Tibet was one of the centers of barley domestication[3–7]. This hypothesis was based on: (1) the discovery of six-rowed wild barleys (*Hordeum agriocrithon* Åberg) in Tibet and surrounding areas[3–8] (Ganzi, Sichuan province, China, Fig. 1); (2) the discovery of barleys with an intermediate phenotype between wild barley (*Hordeum vulgare* ssp. *spontaneum*) and qingke, such as two-rowed hulless barley and six-rowed hulled barley in Tibet[3–5]. *H. agriocrithon* and intermediate barley do not exist as wild populations in Tibet but occur as weeds only at the edges of fields in the region, and have been known as Tibetan weedy barley by Tibetans for generations and described as Tibetan semiwild barley by some barley researchers[3–5]. It should be noted that while Tibetan weedy barley or Tibetan semiwild barley is not a name used in standard barley taxonomy, it has been a popular name used in Tibet by Tibetans or some qingke researchers to distinguish qingke from other Tibetan barleys. Tibetan weedy barley may have made more genetic contribution than Near East barleys to Chinese barleys[6,7]. However, many studies suggested that *H. agriocrithon* may have originated from natural hybridization between *H. spontaneum* and six-rowed domesticated barley[9–11]. Thus the existence of Tibetan wild barley[12] provided only weak support to the hypothesis of Tibet representing one of the centers of barley origin or domestication.

Several routes have been proposed to explain how western Eurasian domesticates such as wheat and barley may have entered East Asia. One of the proposed routes is that wheat and barley entered East Asia from areas to the north of the Tibetan plateau via the Inner Asian Mountain corridor that skirts the Taklimakan desert to the south and the Inner Asian Mountains[13–19] (route I, Fig. 1). Both crops arrive on the northeastern and southeastern Tibetan plateau by 4000 calendar years before the present (cal y B.P.)[19] (route II, Fig. 1). This route of transmission has been favored in archaeological research, but it is also the area in which the most archaeobotanical research has been carried out. Another scenario proposes that these domesticates could have moved east along the southern rim of the Tibetan Plateau: an area where unfortunately little archaeological research has been carried out[20–22]. In sites in the northeastern Tibetan plateau, barley occurs alongside wheat, but also with other crops native to China such as broomcorn and foxtail millet[19]. When barley does appear in central Tibet, and to some extent the southeastern Tibetan plateau, it appears alongside other agricultural products. In addition to Chinese millets, a variety of other southwest Asian domesticates including pea and rye appear at Changguogou in the Yarlung Tsangpo river basin of southern Tibet[22]; and flax at Ashaonao on the southeastern Tibetan plateau[23,24] by roughly 3500 cal y B.P., indicated that the introduction of Tibetan barley from South Asia was also possible. Genetic analysis of barley populations thus might help reveal which routes barley has taken on its spread to the plateau.

Single nucleotide polymorphisms (SNPs) derived from RNA-seq have been used to study the genetic relationship between qingke and other barleys[25,26]. However, most of the samples in these studies were cultivars lacking unambiguous geographic origin information rather than geo-referenced landraces; furthermore, the sample size of qingke in these studies was insufficient (less than 20) to represent the qingke populations. According to an official record (http://www.stats.gov.cn/tjsj/tjgb/ rkpcgb/dfrkpcgb/201202/t20120228_30406.html), by 2010, there are nearly three million people living in Tibet (Supplementary Table 1). Most of them live in southern and eastern Tibetan area. Sixty-nine qingke landraces and 35 qingke cultivars (produced by cross-breeding with different qingke landraces), and ten Tibetan weedy barleys, were collected from the major inhabited areas in seven Tibetan regions and adjacent areas (Qinghai and Yunnan province), to represent the diversity present in Tibetan barley (Fig. 1b; Supplementary Table 1).

Here, we investigate the origin and domestication history of qingke by whole-genome sequencing (WGS) of Tibetan barley including (i) qingke landraces and cultivars from most Tibetan inhabited areas, (ii) Tibetan weedy barleys (including two brittle rachis samples), as well as (iii) eastern and western barley landraces and cultivars (Supplementary Data 1). Population genomic analyses are performed in the context of previously published diversity datasets[27–29] (Supplementary Datas 1 and 2), that comprise barley originating from Africa, Europe, Central, and East Asia including the Tibetan plateau (Fig. 1a). Our analyses strongly suggest that contemporary qingke are derived from eastern domesticated barley providing genomic evidence that the earliest barley was introduced to southern Tibet most likely via north Pakistan, India, and Nepal between 4500 and 3500 cal y B.P.

## Results

**Origin of qingke.** Resequencing of 177 barley genomes, predominantly sampled from Tibet, generated a total of 8.5 terabase (Tb) of high-quality cleaned sequences, with an average of 48.1 gigabase (Gb) per accession (~9.6-fold haploid barley genome coverage, Supplementary Data 1) and revealed 56.3 million (M) SNPs and 3.9 M small insertions and deletions (INDELs) (Supplementary Table 2). A total of 0.54% of the identified SNP and 0.35% of the INDEL polymorphisms resided in coding sequences (CDS) of high-confidence genes[30]. The ratio of nonsynonymous/synonymous SNPs was ~1.11, while 0.23% of the total INDELs led to frameshifts (Supplementary Table 3). An overlapping total of 1.55 M SNPs (Supplementary Table 2) was found in a set of 260 published exome sequences[29] (ES) from a barley world collection.

Using this overlapping set of SNPs in a population structure analyses (principal component analysis (PCA), phylogenomic tree, Fig. 2a, b) separated wild and domesticated barleys into two clades/clusters, which confirmed previous findings[29,31,32]. The domesticated barley clade/cluster was further divided into two subclades/subclusters explained by the geographic origin of the genotypes as reported by Morrell et al.[32]. One clade (clade I, Fig. 2c) included most of the cultivars and landraces of western Asia, central Asia, Africa, and Europe; the other (clade II, Fig. 2c), included landraces of central Asia, eastern Asia, as well as Tibetan barley (qingke and Tibetan weedy barley). For convenience, except the cultivars and Tibetan barley, we defined the domesticated barley landraces of clade I as western barley, and of clade II as eastern barley. Clade II showed that qingke was closer to all eastern than to wild and western barleys.

The evolutionary history of qingke was further inferred by individual ancestry coefficients (Fig. 2d). With *K* from 4 to 9, new subpopulations arose from each barley clade (wild, western, and eastern). PCA confirmed the existence of such subpopulations (Supplementary Figure 1a, c, e). We studied the relationship of these subpopulations with respect to their geographic origin (Supplementary Figure 1b, d, f). For western and eastern barley, accessions originating from close geographic proximity showed a closer relatedness, emphasizing that geographic origin was the main differentiating factor[31,32].

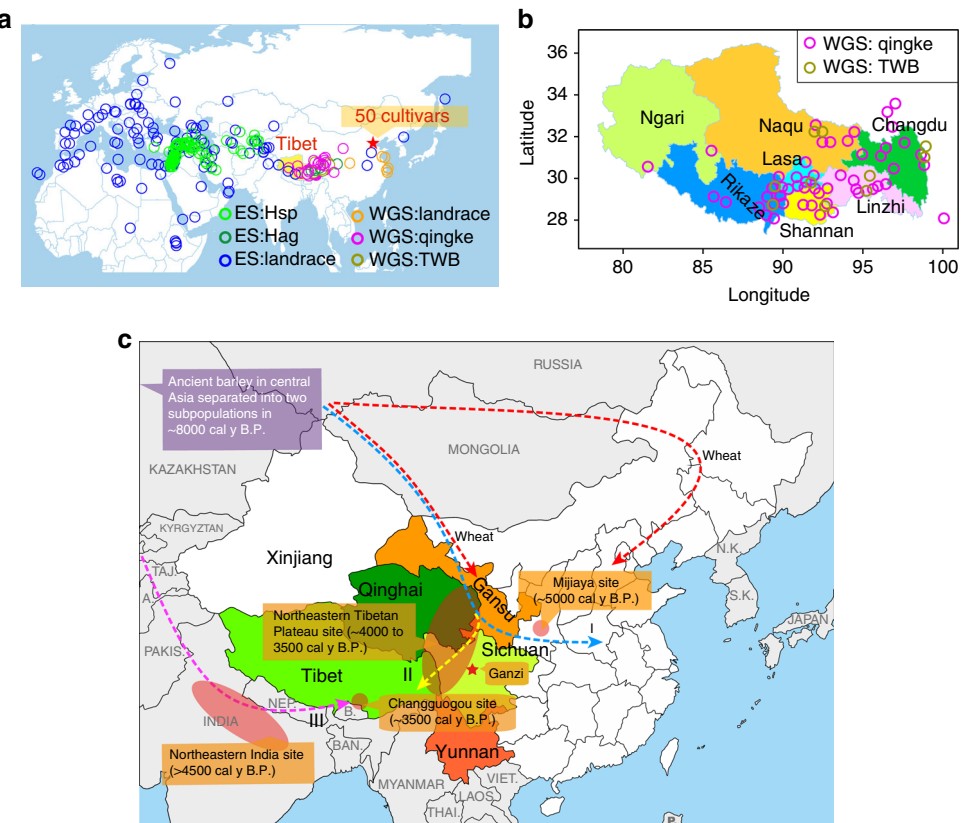

**Fig. 1** Geographical distribution and possible introduction routes. **a** Global distribution of the barley accessions. WGS whole genome sequencing accessions, ES exome sequencing accessions, Hsp *H. spontaneum,* Hag *H. agriocrithon,* TWB Tibetan weedy barley. The pentagram indicates the location of National Crop Genebank of China (NCGC), which provided 50 WGS barley cultivar accessions. **b** Distribution of qingke landrace and Tibetan weedy barleys in seven areas of Tibet. The seven areas are identified by different colors. **c** Possible introduction routes of qingke from Central Asia to Tibet. The nearby provinces (Qinghai, Gansu, Sichuan and Yunnan) of Tibet where Tibetans (Zang people) live are identified by different colors. The circle or ellipse indicates the location of the barley archaeological site, including Mijiaya[18], northeastern Tibetan Plateau[19], northeastern India[21], and Changguogou[22]. The pentagram indicates the location of Ganzi, where ÅEberg discovered and described *H. agriocrithon* in 1938 for the first time[8]. The red dash line indicates the reported introduction route of wheat[13–17]. The blue dash line (route I) indicates the introduction route of barley in North and Eastern China[13–19]. The yellow dash line (route II) indicates the possible introduction route for qingke from Northern Tibet[19]. The magenta dash line (route III) indicates the new discovered route for introduction of qingke from south Asian reported in this study. **a, b** were generated in R (V3.4.3) using packages maps, maptools, plyr, ggrepel, and ggplot2. Geographic information data were obtained from Global Administrative Areas database (GADM V2.8, November 2015). The original map of **c** was from https://d-maps.com/carte.php?num_car = 11572&lang = en. Source data are provided as a Source Data file

By filtering admixed samples, we divided the western and eastern barleys, as well as wild and qingke accessions into four groups based on the PCA: wild, western, eastern, and qingke group (Supplementary Figure 2a). In addition, the majority of the 177 WGS samples, which were qingke and cultivars clustered with western barley (Supplementary Figure 3), were divided into two groups: western cultivars and the qingke group (Supplementary Figure 2b). These defined groups represented the pure barley populations of wild barley, western landraces, eastern landraces, qingke, and western cultivars. Population genetic analysis, including nucleotide diversity ($\pi$), Watterson's estimator ($\theta_W$), gene diversity/heterozygosity ($H_E$), Tajima's *D*, recombination rate ($\rho$), minor allele frequency (MAF) distributions, and linkage disequilibrium (LD) ($r^2$), of the defined barley groups was carried out to test whether Tibet could be recognized as a center of barley diversity and origin as previously suggested[3–7]. This could also help us understand whether it was rather a center of adaptive and diversifying selection. Across the genome, we observed an average reduction in genetic diversity described by $\pi$, $\theta_W$, $H_E$ of 50% in western and eastern landraces relative to wild barley, and ~50% in qingke relative to western and eastern landraces (Table 1; Supplementary Tables 4–6). Interestingly, chromosome 4H exhibited the lowest genetic diversity, which, furthermore,

significantly decreased from wild barley to all domesticated barley groups. Similar observations have been made previously for barley[30] and were also revealed for the syntenic wheat chromosome 4D[33], possibly indicating a universal phenomenon for the group 4 chromosomes of the Triticeae tribe. Overall genetic diversity is higher toward the terminal regions of all chromosomes in barley and low in the proximal nonrecombining regions as shown before[30] (Supplementary Figures 4–7). The wild group had the highest proportion of low-frequency alleles (also supported by the negative Tajima's *D*, Table 1), while the qingke group had the lowest (Supplementary Figure 8). The highest level of LD was present in the qingke group (Supplementary Figure 9), with the genome-wide population recombination rate $\rho$ in qingke estimated to be ~16% of the rate in the western group, or to be ~50% of the rate in the eastern group (Table 1; Supplementary Table 7). Based on the even distribution of the geographic origin of the qingke samples used in this study, representing most of the inhabited area of Tibet, we conclude that the set used is most likely representative for the diversity present in Tibetan barley. The qingke group possessed the lowest genetic diversity, lowest proportion of low-frequency alleles and the highest LD compared to all other wild and domesticated barley groups. All of these factors favor that Tibet was not a center of origin or

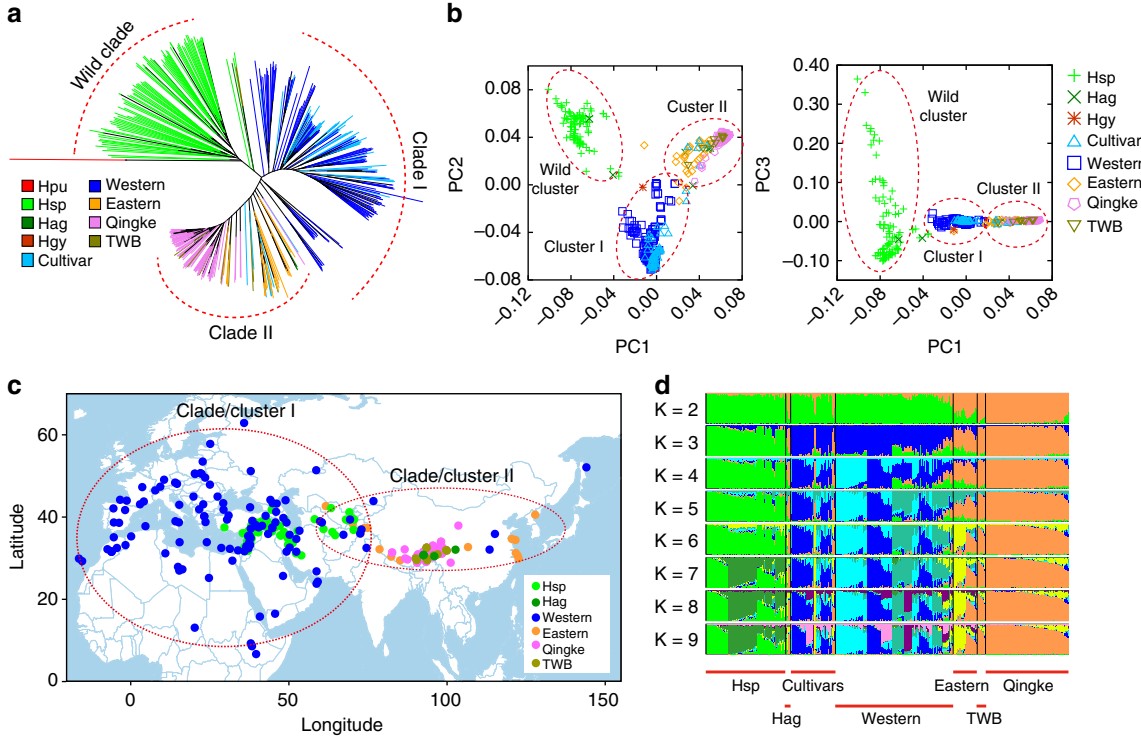

**Fig. 2** Population structure of global barley accessions. Hpu *H. pubiflorum*, Hsp *H. spontaneum*, Hag *H. agriocrithon*, western: landraces in clade/cluster I, eastern landraces in clade/cluster II, TWB Tibetan weedy barley. **a** Neighbor-joining clustering based on genetic distance. **b** Principal component analyses. **c** Global distribution of barley clade/clusters revealed by (**a**, **b**). **d** Individual ancestry coefficients from $K = 2$ to $K = 9$. The red blocks below the plots correspond to the different barley types. **c** was generated in R (V3.4.3) using packages maps, maptools, plyr, ggrepel, and ggplot2. Geographic information data were obtained from Global Administrative Areas database (GADM V2.8, November 2015). Source data are provided as a Source Data file

---

### Table 1 Genetic diversity and Tajima's *D* in barley groups

| | Overlapped SNPs data | | | | WGS SNPs data | |
|---|---|---|---|---|---|---|
| | Wild | Western | Eastern | Qingke | Western cultivars | Qingke |
| $\pi$ | 3.03 | 2.06 | 1.75 | 1.10 | 2.77 | 1.49 |
| $\theta_W$ | 5.39 | 2.07 | 1.57 | 1.00 | 2.79 | 1.47 |
| $H_E$ | 2.85 | 1.94 | 1.52 | 0.99 | 2.51 | 1.36 |
| $\rho$ | 2.85 | 0.23 | 0.07 | 0.04 | 0.32 | 0.06 |
| Tajima's *D* | −1.16 | 0.40 | 1.23 | 0.86 | 0.39 | 0.53 |

Diversity ($1 \times 10^{-3}$) is described by nucleotide diversity ($\pi$), Watterson's estimator ($\theta_W$), gene diversity/heterozygosity ($H_E$), and recombination rate ($\rho$) and reported per bp. Source data are provided as a Source Data file

---

domestication of barley. A very recent study on *H. agriocrithon* came to a similar conclusion[11].

We applied *D* statistics for a better understanding of the relationship between qingke and the other barley groups and to infer the most likely origin of qingke. First, we studied the relationship of the qingke group with the wild, western, and eastern groups (Fig. 3a, b; Supplementary Table 8). The wild group was geographically and genetically divided into two subpopulations of western Asia (wild-WA) and central Asia (wild-CA). With the qingke group fixed in $P_3$, the highest *D* value was generated when $P_1$ was the eastern group, and the second highest appeared when $P_1$ was the western group. Although western domesticated barley (western group) had a greater distance to Tibet geographically than wild-CA barleys originating from central Asia to Tibet, they had a more positive *D* value. This

indicated a domesticated barley origin for qingke, e.g., from eastern domesticated barley.

We also studied qingke in relation to two subpopulations of the eastern group representing Central and South Asian origins like North Pakistan, India, and Nepal/Western Tibetan Plateau (eastern-CA) and East-Asian origins like East China and the eastern Tibetan Plateau (eastern-EA) (Fig. 3c, d; Supplementary Table 9). When $P_3$ was fixed by qingke, *D* value of $P_1$ = eastern-CA was more positive than $P_1$ = eastern-EA, revealing a higher probability for a South Asian introduction of barley into Tibet, which is in contrast to traditional narratives for the introduction of wheat and barley to the Tibetan plateau through Northwest China[13–19]. Our analysis did not provide evidence for the East-Asian introduction of barley into Tibet but favored an introduction via North Pakistan, India, and Nepal to the southern Tibetan plateau (route III, Fig. 1c). This hypothesis was supported by a recent archaeological study[21], which reported some new barley archaeological sites in northeastern India. The newly discovered carbonized barley is earlier (~4500 cal y B.P.) than the previously reported archaeological sites in the northeastern Tibetan Plateau[19] (~4000–3500 cal y B.P).

We observed that genome-wide genetic diversity, presented by per bp value (Table 1; Supplementary Tables 4–7) or by 10 kb windows' value across barley genome (Fig. 4a; Supplementary Figures 4–7), to be lower in qingke compared with other barley groups, indicating a founder effect event (bottleneck) in the history of qingke. Demographic analyses based on our WGS SNPs revealed the effective population size was low in qingke compared to eastern Asian landraces and western cultivars from ~2000 to ~4500 years ago, suggesting a continuous 2500 years' founder effect event (Fig. 4b). We surmised three possibilities to explain the founder effect as follows: (i) A small subpopulation of

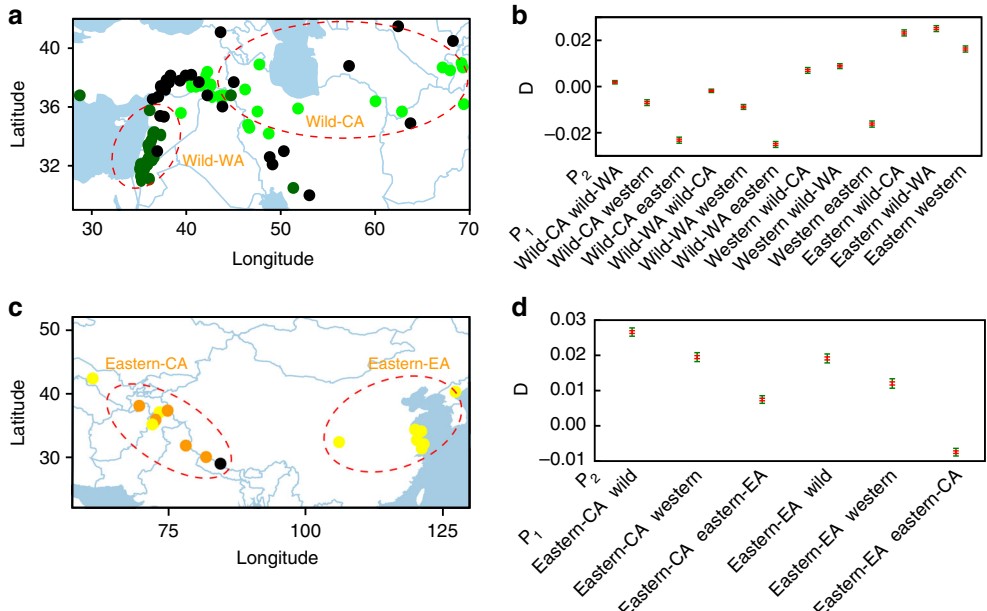

**Fig. 3** Genetic relationship between qingke and other barleys. **a, c** Distributions of the subpopulations of wild barley (**a**) and eastern barley (**c**), respectively, revealed by sNMF and PCA. wild-CA wild barley subpopulation distributed in central Asia, wild-WA wild barley subpopulation distributed in western Asia, eastern-CA eastern barley subpopulation distributed in central and southern Asia, eastern-EA eastern barley subpopulation distributed in eastern Asia. (**b, d**) $D$ statistics for different quadruples of barley populations ($P_1$–$P_3$ and outgroup). Positive $D$ values indicate that $P_1$ shares more derived alleles with $P_3$ than $P_2$ does. Red bars correspond to ±1 standard errors, and green bars correspond to ±3 standard errors. $P_3$ was fixed by qingke, and $P_4$ was fixed by *H. pubiflorum*. **b** $D$ statistics for different comparisons among wild-WA, wild-CA, western, and eastern barley. **d** $D$ statistics for different comparisons among wild barley, western barley, eastern-CA, and eastern-EA. **a**, **c** were generated in R (V3.4.3) using packages maps, maptools, plyr, ggrepel, and ggplot2. Geographic information data were obtained from Global Administrative Areas database (GADM V2.8, November 2015). Source data are provided as a Source Data file

eastern domesticated barley with an initial small effective population was introduced to Tibet and evolved to qingke. (ii) Tibet has complex geographic patterns, including plateau in the west, river valleys in the south, and canyons in the east. The barley which was adaptable to the Tibetan local environment was possibly selected by Tibetan settlers as main local landraces. (iii) A six-rowed spike has more grains than a two-rowed spike, and a hulless caryopsis was more convenient for human food than a hulled caryopsis, resulting in Tibetans preferring the six-rowed hulless barley. Over the same period (~2000–~4500 cal y B.P.), the effective population size value of the eastern Asian landraces, which entered central China around Tibet (route I, Fig. 1c), remained constant between ~4374 and ~4500 (Fig. 4b), indicating the Tibetan Plateau environment provided the main factor resulting in a founder effect. However, the small sample size of eastern barley in central Asia in this study limited the further examination of the founder effect. For instance, we do not know the distribution and proportion of six-rowed hulless barley in eastern barley in general, or whether the six-rowed hulless barley was selected in India and Nepal before introduction to Tibet. Resolving this will require the comprehensive investigation and collection of barley landraces in central and southern Asia.

The postulated beginning of the founder effect (~4500 cal y B.P.) is very similar to the age of ancient barley (>4500 cal y B.P.) in northeast India[21]. Considering barley had arrived in southern Tibet by ~3500 cal y B.P.[22] (Changguogou site), we inferred the approximate earliest introduction time of qingke to southwestern Tibet was between 4500 and 3500 cal y B.P.

The differentiation of two subpopulations for eastern barley was revealed by their individual ancestry coefficients (Fig. 2d; Supplementary Figure 1e, f) indicating that ancient barley in central Asia split into two clades. One spread to India, Nepal, and Tibet evolving into qingke (route III, Fig. 1c); the other entered

North and East China possibly from areas in the northwestern China (route I, Fig. 1c). The demographic analyses revealed the separation time of the two clades was ~8000 cal y B.P (Fig. 4c).

We used the fixation index ($F_{ST}$) approach to investigate the selection signals of local population adaptation for qingke in the exome capture target region of the barley genome (Fig. 5a; Supplementary Data 3). By comparing qingke with eastern landraces, eight regions were identified as candidate selective regions, including the region of *Naked caryopsis* (*nud*) residing on chromosome 7H. Strong selective sweep signals were revealed in several $F_{ST}$-based candidate selected regions (Fig. 5b). The analysis relying on exome capture target regions only did not provide sufficient resolution for determining the exact physical boundaries of the selective sweeps and for identification of individual major selected genes. This step of analysis will depend on the accumulation of additional whole genome resequencing data of eastern barley in future studies.

**Qingke was derived from barley in the Fertile Crescent.** In addition to global barley diversity analysis the haplotype of five genes involved in three key domestic traits of barley was determined. These were the genes *nonbrittle rachis* (*btr1* and *btr2*), *six-rowed spike* (*vrs1* and *int-c*), and the *naked caryopsis* (*nud*)[34–37]. Sanger sequencing confirmed the accuracy of the automated genotype calls for these genes.

Qingke shared the same haplotypes with other domesticated barleys at the *btr1*, *btr2*, and *int-c* loci (Supplementary Datas 4–6; Supplementary Figures 10 and 11). All of the hulless barleys, including qingke and two Ethiopian hulless landraces (WGS-Ld1 and WGS-Ld2), showed the same large 1.6 kb deletion involving the *nud* gene (Supplementary Data 7; Supplementary Figure 12).

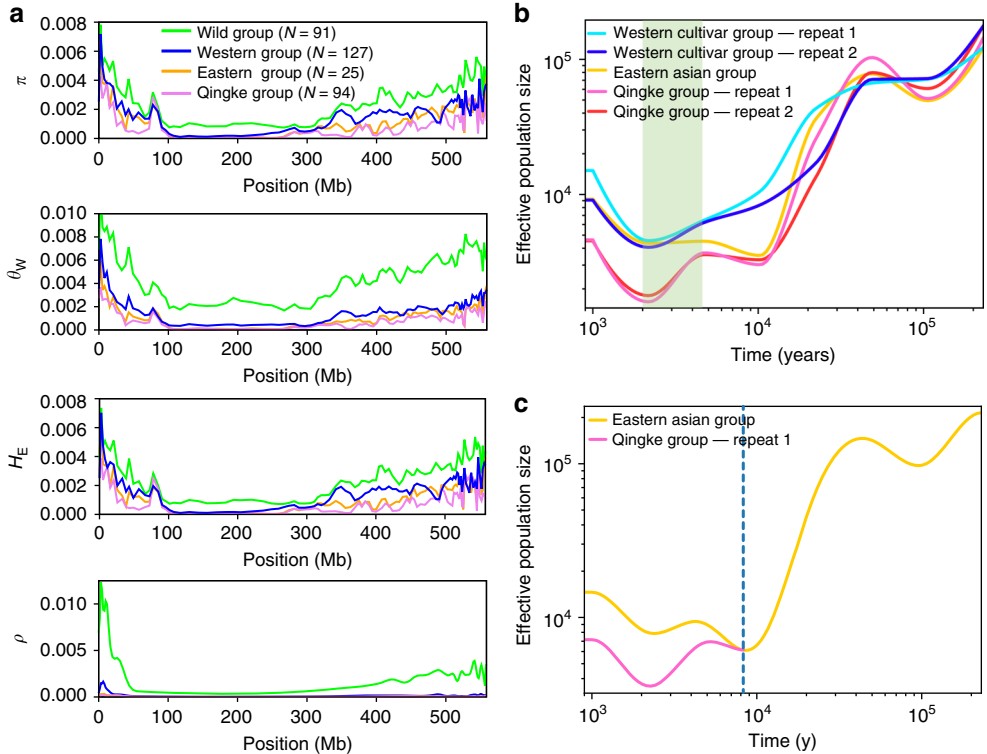

**Fig. 4** Founder effect event of qingke. **a** Genome-wide genetic diversity in chromosome 1H in barley groups. Values of unbiased nucleotide diversity ($\pi$), Watterson's estimator ($\theta_W$), gene diversity/heterozygosity ($H_E$), and recombination rate ($\rho$) were plotted with "smooth bezier" treatment of Gnuplot (http://www.gnuplot.info/). Values on the other chromosomes (2H–7H) are presented in Supplementary Figures 4–7. Qingke showed lower genome-wide genetic diversity compared to other barley groups. **b** Effective population size inferred by SMC++[66] based on WGS SNPs data for qingke, western cultivar, and eastern Asian groups. The SMC++ run was repeated two times with seven randomly chosen accessions of qingke and western cultivar groups, respectively. A marked decline of effective population size of qingke compared with eastern Asian group from ~2000 to ~4500 years ago is highlighted in green. **c** Split time when qingke and the eastern Asian landrace separated from their ancestry inferred by SMC++[66]. The dash line indicates the split time was nearly 8000 years ago. Source data are provided as a Source Data file

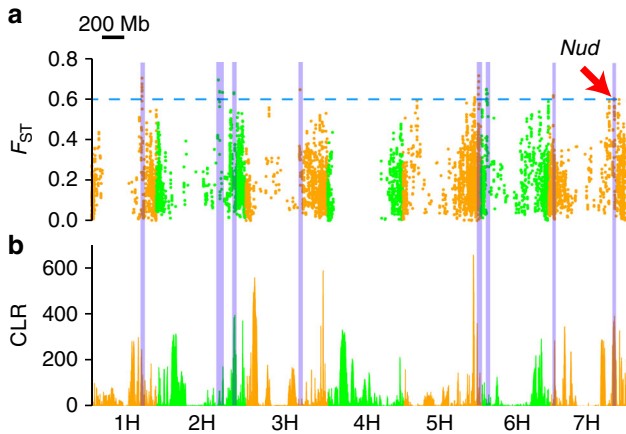

**Fig. 5** Candidate genomic selective of qingke. **a** Hudson's $F_{ST}$ for 10 kb windows compared between qingke and eastern groups. Regions of $F_{ST} \geq 0.6$ above the blue dash line were considered as the candidate selective regions. The *Nud* locus for naked caryopsis was 14 Mb away from a candidate window located at 560.94 Mb in 7H. **b** The selective sweep signal of the qingke group. Large values of the maximum composite likelihood ratio (CLR) correspond to strong sweeps. Candidate selective regions revealed by $F_{ST}$ are highlighted in blue in both (**a**, **b**). Source data are provided as a Source Data file

Except three two-rowed qingke cultivar accessions (WGS-Qk67, WGS-Qk68, and WGS-Qk69), 30% of qingke accessions carried the allele *vrs1.a1*, while other 70% carried the allele *vrs1.a4* reported by Cuesta-Marcos et al.[38] (Fig. 6b; Supplementary Data 8). A recent study[11] suggested *vrs1.a4* arose in a wild barley population of Uzbekistan in central Asia. This underpinned a central Asian origin of qingke and supported our inference that qingke was derived from eastern domesticated barley (Fig. 6d). Mutations in the coding region of *vrs1* converted two-rowed into six-rowed barley[35]. However, six-rowed *vrs1.a4* carriers have not been found to show any lesions within the *Vrs1* ORF[38]. The markedly reduced abundance of the *Vrs1* transcript in *vrs1.a4* carriers has been proposed as the determinant of the six-rowed phenotype, but the causative mutation/s have not been identified[38,39]. The *vrs1.a4* of six-rowed qingke showed no mutation in any exon of the gene as reported[38] with the only unique SNP being at bp-position 387 of the *vrs1.b* complete gene sequence (gi|119943316), therefore, residing in an intron. The SNP has been reported in four *vrs1.a4* accessions[38], and was found uniquely fixed in 87 *vrs1.a4* carriers in this study, while other reported *vrs1.a4* variants[38] were not uniquely fixed (Fig. 6a). Thus the SNP might be a key potential mutation causing the six-rowed phenotype and should be studied in the future.

Altogether, qingke shared the known domestication gene alleles with worldwide cultivated barley thus supporting that barley domestication has only occurred in the Fertile Crescent[11] and rejecting a Chinese (or Tibetan) domestication of qingke.

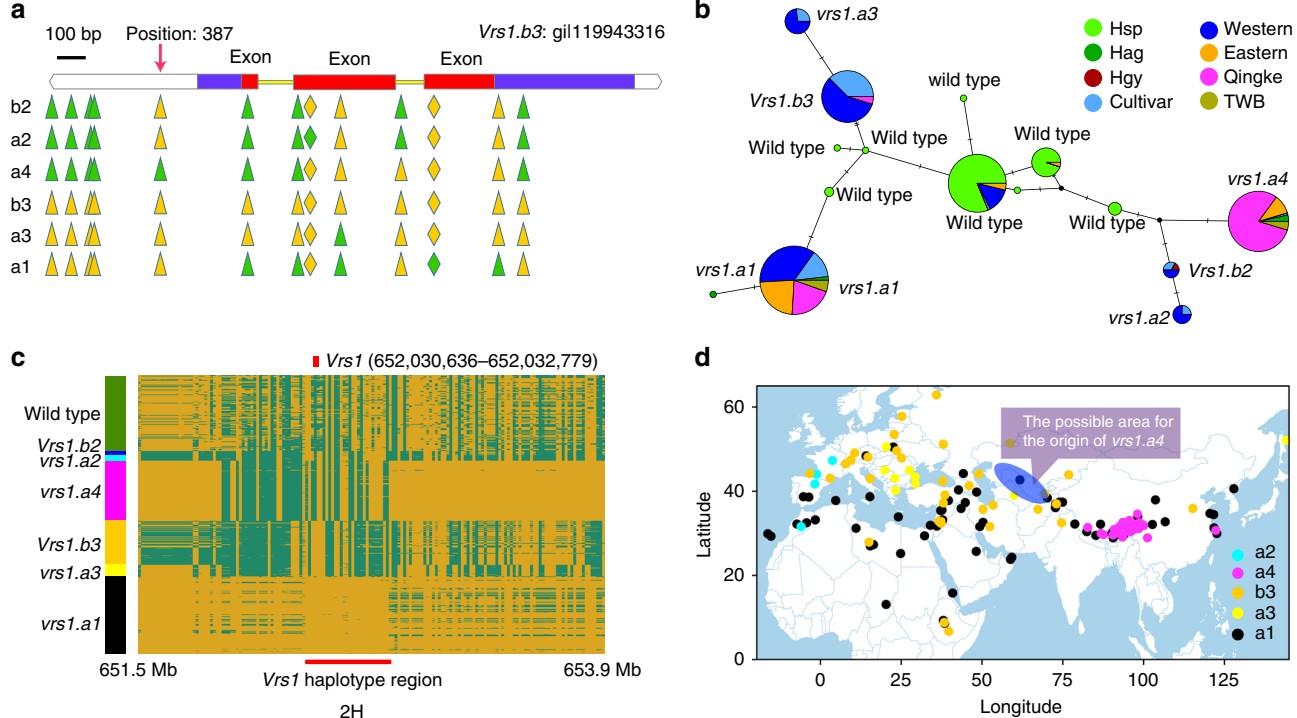

**Fig. 6** Molecular and spatial variants in *Vrs1*. **a** Gene structures (exon: red bar; intro: yellow bar; UTR: blue bar) of *Vrs1* with the relative positions of the SNPs (triangle) and INDELs (rhombus), respectively. The golden triangle or rhombus indicates the same genotype compared with the reference gene sequences (*Vrs1.b3*: gi|119943316) and the lime green indicates difference. The *vrs1.a4* alleles differ from other domesticated *Vrs1* alleles by a unique SNP (G/T) in position 387 of the *Vrs1.b3* sequences. **b** Median-joining networks *Vrs1* haplotypes of barley accessions based on *Vrs1* SNPs. The INDELs were considered as SNPs for distinguishing b2 and a2. Hsp *H. spontaneum*, Hag *H. agriocrithon*, Hgy *H.* var. *gymnospermum Korn*, TWB Tibetan weedy barley. Three two-rowed qingke cultivar accessions which mixed with pedigree of two-rowed barley, carried the allele *Vrs1.b3*. **c** Degree of haplotype diversity around the *Vrs1* locus in the barley genome. The golden region of corresponding accessions indicates the same genotype compared with the barley genome (Morex: *Vrs1.a1*); and the green region indicates difference. **d** Haplotype distributions according to geography for *Vrs1*. The six *Vrs1.b2* accessions were not shown on the map because of the absence of clear geographical location. Uzbekistan highlighted by blue ellipse was the possible origin area of *Vrs1.a4* reported by Pourkheirandish et al.[11]. **d** was generated in R (V3.4.3) using packages maps, maptools, plyr, ggrepel, and ggplot2. Geographic information data were obtained from Global Administrative Areas database (GADM V2.8, November 2015). Source data are provided as a Source Data file

**Origin of *H. agriocrithon* and Tibetan weedy barleys**. The origin of Tibetan wild or weedy barley was still unclear despite the previous studies[8–12]. Six *H. agriocrithon* (Tibetan six-rowed wild barley) accessions and ten Tibetan weedy barley accessions were included in this study for resequencing. In all analyses of population structure (phylogenetic tree, PCA, and individual ancestry coefficients) only two *H. agriocrithon* accessions clustered close to the true wild barley (*H. spontaneum*) cluster (Fig. 2a, b); all other accessions of *H. agriocrithon* and Tibetan weedy barleys clustered within the eastern barley as previously reported[11,34,40]. Although two of the six *H. agriocrithon* accessions in the present study clustered with *H. spontaneum*, at the six-rowed trait locus *vrs1* all *H. agriocrithon* accessions showed the six-row conferring haplotypes *vrs1.a1* or *vrs1.a4* indicating their feral or hybridization origin from domesticated or partially domesticated barley[8–11] (Fig. 6). Ten Tibetan weedy barleys showed slightly higher genetic diversity ($\pi = 1.43 \times 10^{-3}$, $\theta_W = 1.37 \times 10^{-3}$) than qingke (Table 1), but lower than other domesticated barleys (Table 1), and also showed domestication gene haplotypes that were shared with domesticated barley. At the *btr1* locus, we found two Tibetan weedy barleys (WGS-Tw4 and WGS-Tw7) with brittle rachis. They exhibited a new haplotype which could have occurred from hybridization between the western (*btr1Btr2*) and the eastern domesticated type (*Btr1btr2*) followed by recombination between the *btr1* and *btr2* loci (Supplementary Datas 4 and 5; Supplementary Figure 10). Such a recombination could have restored the *Btr1Btr2* genotype conferring brittleness of the rachis thus

mimicking the phenotype of true wild barley. In a recent report by Pourkheirandish et al.[11], the same scenario was proposed for Tibetan *H. agriocrithon* having originated from hybridization between six-rowed landraces carrying *btr1Btr2* and *Btr1btr2* genotypes.

Qingke was the dominant barley in Tibet and other Tibetan barley were considered as weeds by Tibetans in their agricultural activity for generations[3–5]. These weeds might have occurred through fertilization on field borders and spread of seeds through animal dung. In addition, a similar example was reported for weedy rice explained by de-domestication[41,42]. Although *H. agriocrithon* and other Tibetan weedy barley accessions were only present as a small sample in this study, our results, without exception, supported clearly feral or hybridization origin of Tibetan wild barley[11].

## Discussion

We want to conclude with a cautionary note on methodology. Our conclusions regarding the origin of qinqke are supported by multiple lines of evidence from analysis carried out at the whole-genome level. However, there are multiple of reasons why caution needs to be applied at understanding patterns of diversity at finer scales: (i) incompleteness and inaccuracy of the current barley reference genome sequence assembly[43]; (ii) uncertaties of aligning short reads in a plant species with a large and complex reference genome, which may result in reduced mapping rates for

haplotypes divergent from the Morex reference; and related to this (iii) presence–absence variation, which results in ignoring sequence variation in genes absent from the Morex reference. In the future, the construction of multiple high-quality reference sequences including representatives of qingke germplasm may contribute to obtaining a full picture of haplotype diversity in barley.

Our population genomics study showed that qingke originated from the eastern domesticated barleys. Qingke landraces brought to the Tibetan plateau were derived primarily from south Asian origin between ~4500 to ~3500 cal y B.P., supporting the hypothesis that a southern route of crop introduction into Tibet was an important factor. A founder effect event occurred in the qingke population between ~4500 to ~2000 cal y B.P. The accumulated sequence data as well as the SNP and INDEL information will be of great use for further barley population genomic studies.

## Methods

**Sample preparation and sequencing.** The 172 WGS barley accessions used in this study, including wild barleys (including a semiwild accession: *Hordeum vulgare var. gymnospermum* Korn), cultivars (produced by cross-breeding), western and eastern landraces, qingke landraces, qingke cultivars (produced by cross-breeding with different qingke landraces; three two-rowed qingke accessions, including WGS-Qk67, WGS-Qk68, and WGS-Qk69, were produced by the cross-breeding between qingke landrace and two-rowed domesticated barley), Tibetan weedy barley, were provided by (i) National Crop Genebank of China (NCGC), and (ii) Tibetan Academy of Agricultural and Animal Husbandry Sciences (TAAAS) (Supplementary Data 1). NCGC is an official germplasm resource bank of Chinese Academy of Agricultural Sciences and the detailed information of the barley accessions was available on the website http://www.cgris.net/cgris_english.html. The DNA was extracted from 4 week old seedling's leaves. The sequencing was performed on an Illumina Hiseq2000 or Hiseq4000 platform.

Previously published WGS samples, including *Hordeum pubiflorum*[28], 5 barley cultivars[27]; and 260 exome sequencing samples[29], including 97 wild barleys accessions (91 *H. spontaneum* and 6 *H. agriocrithon* accessions) and 163 landraces, were downloaded (Supplementary Data 2).

**Alignment and variant calling.** We cleaned the Illumina NGS raw data to remove adaptors, trim low-quality bases and also to remove "N" with Trimmomatic[44] (V0.36). The clean reads were mapped to the barley genome reference[30] with BWA[45] (V0.7.10-r789, mapping method: MEM). The high-quality mapped reads (mapped, nonduplicated reads with mapping quality ≥ 20), were selected with Samtools[46] (V1.3.1) commands "−view −F 4 −q 20" and "−rmdup". For 178 WGS samples (including *Hordeum pubiflorum*), the mapping statistics based on high-quality mapped reads of each accession included: (1) the coverage depth of each chromosomal position (Samtools command "−depth"); (2) proportion of barley genome covered by different read depths (Supplementary Figure 13a). Considering there were average ~30% uncovered region of barley genome for WGS accessions (Supplementary Figure 13b), the region covered by at least two reads in ≥80% of the WGS accessions were defined as the WGS effective covered region of barley genome. The overlap between WGS effective covered region and exome target region[29] (https://doi.org/10.5447/IPK/2016/27) were called as overlapped effective covered region (Supplementary Table 10).

Only high-quality mapped reads were used for variants calling. BAM files were sorted and marked PCR duplication by Picard (V1.117, http://broadinstitute.github.io/picard/), then variants calling was performed using the Genome Analysis Toolkit[47] (GATK, V3.3-0-g37228af). The 178 WGS and 260 ES samples were combined for variant calling. For barley no whole genome SNP and INDEL datasets was available to carry on Base Quality Score Recalibrator (BQSR) and INDEL Realigner, so we used the following approach as GATK website recommend for non-human data (https://gatkforums.broadinstitute.org/gatk/discussion/1706/best-recommendation-for-base-recalibration-on-non-human-data). First, we did an initial round variants calling for our original data by round HaplotypeCaller. Without available truth/training variants, we used GATK tool harder filter to filter the variants instead of Variant Quality Score Recalibration as GATK recommend (https://software.broadinstitute.org/gatk/documentation/article.php?id = 3225). The parameters of hard filter was set by default (for SNPs: QD < 2.0, FS > 60.0, MQ < 40.0, MQRankSum < −12.5; for short INDELs: QD < 2.0, FS > 200.0, ReadPosRankSum < −20.0). Applying the hard filter provided an initial confidence in the SNPs and INDELs sets. Second, the original BAM files were treated by BQSR and INDEL Realigner using the initial confidence SNPs and INDELs. Using of HaplotypeCaller and hard filter again, further improved confidence in the SNPs and INDELs sets. This dataset including 438 accessions (including 1 *Hordeum pubiflorum*, 177 WGS and 260 ES barley accessions) was considered as the raw confidence variants sets.

To obtain high-quality variants sets, the raw confidence variants were filtered on the basis of the steps Russell et al.[29] used for barley exome resequencing data. For WGS data, the variants of 177 WGS barley accessions were extracted from raw confidence variants sets and performed the following filtering steps: (1) only variants in the WGS effective covered region were kept; (2) only bi-allelic and polymorphic variants were kept; (3) genotype calls were considered successful if read depth was ≥2 and ≤50, otherwise were regarded as missing; (4) variants positions with more than 80% heterozygous calls or more than 20% missing genotype calls were discarded; (5) both alleles of a variant were required to occur in at least one individual in the homozygous state. For the overlapped variants between WGS samples and ES samples, the variants of 437 barley samples (without *Hordeum pubiflorum*) were extracted from raw confidence variants sets and performed the following the similar filtering steps showed above. The differences were: (1) only variants in the overlapped effective covered region were kept and (2) genotype calls were considered successful if read depth and the genotype quality score were both ≥10 for deeply sequenced ES samples as Russell et al.[29] used.

After filtering, two kinds of high-quality variants were obtained. One was the genome-wide variants (the SNPs data were called WGS SNPs data in the following analyses) of 177 WGS barley accessions; the other was the overlapped SNPs between the 177 WGS accessions and the 260 ES accessions (the SNPs data were called overlapped SNPs data in the following analyses). For estimating the quality of our genome-wide SNPs, ten primers were designed by Primer-BLAST (https://www.ncbi.nlm.nih.gov/tools/primer-blast/index.cgi?LINK_LOC = BlastHome). Thirteen accessions, including two wild barleys, six cultivars, and five qingke accessions were used for performing Sanger sequencing (Supplementary Datas 9 and 10). The annotation of genome-wide SNPs and INDELs were based on barley's high-confidence (39,734 genes) and low-confidence (41,545 genes) gene sets[30] with an in-house Perl script and Reseqtools[48] (V0.25, https://github.com/BGI-shenzhen/Reseqtools).

**Population structure analyses.** Phylogenomic tree was constructed on the basis of the distance matrix calculated by the software PHYLIP 3.68 (http://evolution.genetics.washington.edu/phylip.html), and presented by iTol[49] (V3, http://itol.embl.de/). PCA was performed with EIGENSOFT[50] (V6.0.1).

The individual ancestry coefficients of overlapped SNPs data was performed with sNMF[51] (V1.2), which was more appropriate to deal with inbred species[51]. Moreover, we choose sNMF because it had shown good performance in barley[29]. Before running the sNMF, we treated the overlapped SNPs sets as haploid, coding all heterozygous sites as missing data. The command sNMF was called with parameter −m 1 (assuming haploid data for the predominantly inbreeding barley) for K values between 1 and 15. For each K value, 100 replications runs were performed with random, varied seed. The Q proportions were averaged across the 10 replications with the lowest cross-entropy by CLUMPP[52] (V1.1.2) and plotted by Distruct[53]. The K = 9 was chosen because the major subpopulations reported by Russell et al.[29] occurred from K = 2 to K = 9 and these stable subpopulations results were enough for the following analyses. For wild and western barley, if the components for subpopulations within the major group was ≥0.65, the samples were classified as wild or western, respectively, and the remaining samples were deemed admixed. For eastern barley, the critical components value was set 0.5. For proved the existence of theses subpopulations, we performed the PCA for wild, western and eastern barleys with EIGENSOFT[50] (V6.0.1), respectively.

**Population genetic statistics.** Only the non-admixed samples in both PC1–PC2 and PC1–PC3 of PCA results were defined as groups (Supplementary Figure 2). Admixed samples can be the result of recent outcrossing between traditional qingke and Chinese elite varieties growing side-by-side in the fields of Tibetan farmers. Such admixed samples would not be informative about the origin and history of qingke. The groups with sample size ≥20 were used in the following population genetic analyses.

The heterozygous SNPs proportion in the total SNPs was <2% in most of the accessions, meanwhile the accessions with high heterozygous SNPs proportion (>4%) comprise a large proportion of cultivars, indicating barley is a tightly inbred species (Supplementary Figure 14). Inbred samples would seem to be closer to haploid than diploid[54], for all of the following population genetics analyses, we treated SNPs datasets as haploid, coding all heterozygous sites as missing data. LD was calculated using PopLDdecay[55] (V3.31) with command "−MAF 0.01 −Het 0.8 −Miss 0.8 −MaxDist 1000" in barley groups. Regarding the LD for overall genome, the pairwise $r^2$ value was calculated for individual chromosomes using SNPs from the corresponding chromosome and then the pairwise $r^2$ values were averaged across the whole genome. The nucleotide diversity ($\pi$), Watterson's estimator ($\theta_W$), gene diversity/heterozygosity ($H_E$) per bp and Tajima's D for haploid data were estimated with our in-house Perl scripts based on their definitions[56–59]. The unbiased value of a window for $\pi$, $\theta_W$, and $H_E$ was equal to the sum of the value per bp divided by the corresponding effective covered region size of the window. The recombination rate ($\rho = 4N_e r$) was estimated using a composite likelihood approach[60] with Maxhap (http://home.uchicago.edu/rhudson1/source/maxhap.html) on a per-contig level[29,61]. We considered only SNPs with minor allele counts ≥3 located in contigs that contained at least 20 SNPs. Values of $\rho$ per bp were estimated across a grid of values from $1 \times 10^{-4}$ to 0.2, assuming no homologous gene conversion. The fixation index ($F_{ST}$) between pairwise groups was calculated

using Hudson's estimator with the explicit formula given as Eq. (10) in Bhatia et al.[62] (Supplementary Table 11), which were independent of sample sizes. The average $F_{ST}$ of a window was considered when the windows comprised at least 5 SNPs per kb. For windows calculation of $\pi$, $\theta_W$, $H_E$ and $F_{ST}$, the windows size was set as 10 kb with 2 kb step. Only the windows which comprised ≥2 kb effective covered region were considered. The distribution of $\pi$, $\theta_W$, $H_E$, and $\rho$ across the barley genome chromosomes was plotted using Gnuplot (V5.2, http://www.gnuplot.info/) with "smooth bezier" treatment based on the value of per window or per contig.

In addition, we considering ten Tibetan weedy barleys as a group, calculating the nucleotide diversity ($\pi$) and Watterson's estimator ($\theta_W$) based on the overlapped SNPs data using the same methods above.

**D statistics.** The $D$ statistics[63] of four-population were calculated using ADMIXtools[64] (V4.1). The SNPs matrix converted to EIGENSOFT format using fcGENE[65] (V1.0.7) and CONVERTF[64]. The barley relative $H.$ $pubiflorum$ was set as the out-group. The genotype of $H.$ $pubiflorum$ was directly extracted from the raw confidence variants sets of 438 samples.

**Demographic history.** The sequential Markov coalescent implemented in SMC++[66] (V1.13) was used to estimate the demographic history for qingke. The SMC++ was more suit for genome-wide SNPs than exon-wide SNPs[66], thus the three groups (western cultivar group, qingke group, and eastern Asian group) based on WGS SNPs data were used. The population size for SMC++ run was recommend as 2–10 (https://github.com/popgenmethods/smcpp). The eastern Asian group only included seven samples. For normalizing the population size, we randomly selected seven different samples, which evenly distributed in the PCA of western cultivar and qingke groups (Supplementary Figure 15). For each group, two replicated selections were performed. The non-WGS effective covered region was masked with parameter "vcf2smc −m". For each group, every sample was set as the pair of distinguished lineages once (parameter "vcf2smc −d") for generating varied independently evolving sequence. All of the sequences were used as input for each group when running "SMC++ estimate". The split time between qingke and eastern Asian group was estimated by command "SMC++ split". A mutation rate of $6.5 \times 10^{-9}$ per site per generation, which were used for the demographic estimation for rice[67], and a constant generation time of 1 year was assumed to translate coalescence generations into times.

**Candidate genomic region for plateau adaption of qingke.** To examine local population adaptation of qingke, the $F_{ST}$ we calculated above between qingke group and eastern group was used. The windows (10 kb with 2 kb step) in which $F_{ST} \geq 0.6$ were regarded as the candidate selective regions. The genes overlapped with these regions (up and down stream ± 2kb) were considered as candidate genes. These genes were aligned with Swiss-Prot Protein Sequence Bank (Uniprot/release-2015_04) by BLAST[68] (V2.2.26). The $E$-value was set as $1 \times 10^{-5}$. The symbols of Swiss-Prot were used in searching for the gene's function. In addition, Sweep-Finder[69] was used to examine if these candidate regions were overlapped with selective sweep.

**Haplotype of key domestication genes.** The sequences of the five genes ($btr1$, $btr2$, $vrs1$, $int-c$, and $nud$, Supplementary Data 11) were downloaded from NCBI and aligned with the barley genome using IPK's blast server (http://webblast.ipk-gatersleben.de/barley_ibsc/). The genotypes of the five genes were identified as the following approaches. (1) If the gene was annotated in barley genome high-quality gene set ($int-c$: HORVU4Hr1G007040.1; $nud$: HORVU7Hr1G089930.5), we directly extracted its genotype from our high-quality SNPs and INDELs data. (2) If the gene was not annotated in barley genome or not uniquely mapped to barley genome ($btr1$ and $vrs1$), the best mapped region of ±20 kb in barley genome was cut to be used as a reference for variants calling. The variants calling steps were the same as we used in our confidence variants' calling. (3) An assembly error in barley genome resulted in part of $brt2$ sequence mapping to one position, and another part mapping to another position (Supplementary Data 11). We directly used the downloaded wild $Btr2$ sequences (gi|914342917) as a reference for variants calling. The coding region of $btr1$, $btr2$, $int-c$, and $nud$ were not covered by exome sequences, so the haplotype of these four genes were only identified by the 177 WGS samples. The $vrs1$ locus was covered by exome sequences, providing the haplotype in all of the barley samples. Only variants with a MAF ≥ 5% were considered. For estimating the genotype quality of these genes, four primers were designed by Primer-BLAST (https://www.ncbi.nlm.nih.gov/tools/primer-blast/index.cgi?LINK_LOC = BlastHome) for $btr1$, $btr2$, $vrs1$, and $int-c$ (Supplementary Data 11). Seventy-third successful PCR-based Sanger sequences confirmed the genotypes examined. The median-joining haplotype networks based on the SNPs of these genes were constructed with PopART[70] (V1.7). The deletion around $nud$ locus of each sample were revealed by the reads depth. In addition, phased SNPs around each gene locus were prepared using SHAPEIT[71] (v2.r790).The genotypes which were the same as the barley genome, were converted to 0; while the altered ones were converted to 1. The phased two haplotypes of each accessions were plot with Gnuplot (V5.2, http://www.gnuplot.info/).

**Reporting summary.** Further information on experimental design is available in the Nature Research Reporting Summary linked to this article.

**Code availability.** The in-house Perl and Shell scrips for reads mapping, variants calling, filtering and annotation, and population genetic analyses ($\pi$, $\theta_W$, $H_E$, Tajima's $D$, and $F_{ST}$) are available in https://sourceforge.net/projects/origin-of-qingke-barley/files/?source=navbar

## Data availability

All data and genetic material used for this paper are available from the authors on request. The sequences data that support the findings of this study have been deposited in NCBI under the BioProject PRJNA417220 with the Sequence Read Archive (SRA) number SRP131710. The accessions code of the 200 Sanger sequences deposited in NCBI are MG879031 to MG879230. Genotype matrices for SNPs and INDELs are available from https://doi.org/10.5447/IPK/2018/15. Plant materials used in this study can be requested from the corresponding author Nyima Tashi or from the Chinese Crop Germplasm Resources Information System (http://www.cgris.net/). A reporting summary for this article is available as a Supplementary Information file. The source data underlying Figs. 1–6, Table 1, Supplementary Figures 1–9, 10a–c, 11–15, and Supplementary Tables 4–7 and 11 are provided as a Source Data file.

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

## Acknowledgements
We thank Dr. Eviatar Nevo, Dr. Takao Komatsuda, Dr. Jade d'Alpoim Guedes, and Dr. Mark S. Aldenderfer for critical reading of this manuscript and giving important suggestions. We thank State Key Laboratory of Agricultural Genomics (No. 2011DQ782025) for their suggestion. This work was supported by the following funding sources: the Financial Special Fund (2014CZZX001 and 2015CZZX001, 2017CZZX001, and XZNKY-2018-C-021), the Tibet Department of Major Projects (XZ201801NA01).

## Author contributions
N.T., X.Z., and Q.X. designed and managed the project. Y.G. performed all of the bioinformatics analyses and wrote the manuscript. N.S. gave insightful instructions on the data analysis, suggestions, comments, and revision on the manuscript. M.M. gave insightful instructions on the data analysis. N.T., X.Z., S.L., L.M. and R.J.H. revised the manuscript. N.T., X.Z., G.G., H.Y., Y.W., Z.W., S.Z., Y.T. and J.Z. collected the accessions. Q.L., Z.X. and J.Z. helped to plot the figures. N.T., X.Z, S.T., L.S., S.L., L.B., Z.Z., W.H., S.Z., X.F., Q.G., Y.Y., J.W. and H.Y. helped with language editing.

## Additional information

**Journal Peer Review Information:** *Nature Communications* thanks the anonymous reviewer(s) for their contribution to the peer review of this work. Peer reviewer reports are available.

