## [Peer Review File · Nature Communications]

Reviewers' comments:

Reviewer #1 (Remarks to the Author):

Zeng et al. report a study of nucleotide sequence diversity in Tibetan barley. The primary aim is to determine the route of introduction of barley, a staple crop at high elevations in Tibet. The study involves a truly impressive amount of data. There are various numbers listed in the text, but there are ~180 whole genome sequences from barley that appear to be newly reported here. The authors indicate that there have been a number of prior studies of the origin of domesticated barley in Tibet. One issue has been that apparently wild or feral forms of the progenitor species occur within the region, making the origins of the local cultivated form unclear. The authors seem to largely accomplish their aim of determine the origin of Tibetan barley. However, as noted below, there are some quite fundamental issues that should be addressed.

First, for a paper that is almost entirely population genetic analysis, there are a number of material errors in the analysis. The most fundamental issue is with treatment of sample size. The authors don't report heterozygosity estimates, but a little Google searching suggests barley is typically highly inbred. VCFTools, that the authors used for diversity estimates, only reports Watterson's theta and pairwise diversity for diploid samples, and does not account for inbreeding. Pairwise diversity is not a simple function of sample size; you can't calculate the diploid measure and multiply by two! Sample size issues likely also impact estimates of allele frequency divergence (F_{st}) and principal component analysis. In a similar vein, the ADMIXTURE tool used for genetic assignment makes a Hardy-Weinberg assumption. Be sure to check the specific comments below for the lines in the manuscript where these issues with analysis occur in the text.

Second, there are several errors in the presentation of population genetic results. For example, the authors conflate founder effects and selection when discussing the relative diversity in cultivated barley. Founder effects are demographic events and impact diversity genome-wide (see the Wikipedia article). Selection for adaptation tends to act on individual loci. In a related manner, comparisons on allele frequency divergence are presented as "selective sweep" analysis. While in some scenarios, a selective sweep could result in F_{st} differences, more generally F_{st} differences result from long-term differential selection contributing to local adaptation, a type of selection that is quite distinct from a "selective sweep." Be sure to check the specific comments below for the lines in the manuscript where these issues of presentation occur.

Third, the figures could be improved dramatically. For example, the haplotype diagrams convey less information than could be portrayed in a breakdown of haplotypes and their occurrence among accessions.

Fourth, all UNIX shell code used for sequence read mapping, SNP filtering, etc. should be made publicly available in order to increase reproducibility of the work. Github, BitBucket, or a similar online concurrent versioning system is appropriate. I also did not see a statement with regard to the availability of the accessions studied, but hopefully those are also being made availability to barley researchers.

Minor comments:

Line 56: Founder effect and selection are somewhat different phenomenon.

Line 113: A Google search suggests that genetic evidence of Eastern and Western domesticated barley traces to a Michael Clegg paper. Should that be cited here?

Line 131: Comparisons within species aren't really phylogenetic analyses.

Line 140: Again, a google search suggests most references to Eastern and Western cultivated barley turn up Michael Clegg papers. I wonder if they should be cited here?

Line 167: There are a lot of different types of "population genetic analysis." Perhaps be more specific? The following sentences are about comparisons of nucleotide sequence diversity? Is that what the authors are referring to?

Line 171: The relative high nucleotide diversity estimates again sent me back to the literature. These numbers seem similar to, though lower than, those from the 2014 Michael Clegg paper from J of Heredity.

Line 185: Parametric estimates of recombination rate would be preferable here and would more directly parallel the nucleotide diversity estimates. See LDhat for example.

Line 195: There are a half dozen different D statistics in population genetics (Lewontin's D for measuring LD, Tajima's D for site frequency spectrum, etc.). Specify here what is being measured and cite the source of the test, not just the software implementation.

Lines 247-253: More discussion around the three possible scenarios brought up here is needed and interpretation of results relating to these three scenarios.

Lines 274-277: Stronger evidence is needed to claim that the gene HvCRF1 is associated with a selective sweep. Comparing to previously reported genes in other plant species is not strong enough evidence considering all of the genome sizes of the other plants are much smaller than barley. Proportionally, a 14 Mbp stretch is about the same as the other genomes mentioned. Perhaps use SweepFinder (Nielson et al 20015) or XP-CLR (Chen et al 2010) to provide more evidence for this conclusion.

Line 252: The size of the founding population and the strength of selection are likely two separate issues that are being confounded in this presentation. Demography (founder events) tends to impact the whole genome while adaptation tends to cause selection to act on individual loci (Cavalli-Sforza 1966). So in general, adaptation does not impact the whole genome through founder effect.

Line 320: The statement about the Tibetan population at the beginning of the Methods seems out of place.

Lines 354-357: Include logic on using default hard filtering parameters for SNP and indel sets, especially since this study heavily relies on high confidence SNP calls.

Line 430: It is not clear that there is a biological rationale for calculating diversity only for individuals from a region that show no evidence of admixture. Of course, the diversity in a sample owing to admixture remains an important portion of overall diversity.

Line 447: VCFtools calculates θ_{pi} for diploid samples. Inbred barley samples would seem to be closer to haploid than diploid. See Nordborg & Donnelly 1997 in Genetics. This means the number of chromosomes samples is off by a factor of 2 (in the denominator) in any estimate. Both LD and F_{st} estimates are also dependent on sample size.

Line 455: The θ_{pi} estimates are already a parametric estimate. They are adjusted for sample size. Ten sampled chromosomes is generally a sufficient sample size for this type of comparison. See Pluzhnikov and Donnelly 1996 in Genetics.

Line 462: There are 4 or 5 D statistics in population genetics. Cite the original source of the

approach along with the software implementation.

Line 472: Perhaps what the authors want to say here is that “qingke was not directly derived from wild barley”

Line 475: This type of demographic event is not the same phenomenon as a selective sweep. Selective sweeps are locus-specific phenomena. Check the Wikipedia article at: https://en.m.wikipedia.org/wiki/Selective_sweep. In general, it isn't clear what is going on in this analysis. Why limit to the “diversity in top 20%”? Also, again Fst outliers and selective sweeps may be indicative of quite distinct evolutionary processes.

Reviewer #2 (Remarks to the Author):

The paper makes an interesting and timely contribution in an important area of research. The observations here are largely about clarity, and optimal connectivity between text, figures and tables, in both main and supplementary sections.

1. The tables illustrated in the file 'Supp figs with captions' do not match those in the file 'Qingke tables_162380_0_supp_2868546_p66v17'. Perhaps these tables are all meant to be in supplementary data. Clarification needed

2. The titles/legends for the Supplementary Tables (in file 'Qingke tables_162380_0_supp_2868546_p66v17') are inadequate - there is only a brief one-line description in the top row of the relevant Excel sheet, and sometimes some more data at the bottom of the table. Proper legends are needed with fuller explanations.

3. The grammar needs improving in the legends. The grammar is OK in the main body of the paper.

4. Table S1 – This requires accession status (landrace, cultivar, etc), more details of collection sites, including latitudes and longitudes, where these were available.

5. The 'Results' section would be more appropriately titled 'Results and Discussion'

6. 'Tibetan weedy barley' as a category lacks clarity - different morphological types are combined within this category – cf. lines 64 to 84, and 110 to 116 of the introduction, and then the section 'Origin of *H. agriocrithon* and Tibetan weedy barleys' in Results. It would be helpful to the reader to separate these types out and call them something different - not necessarily in the analyses, but in the discussion of results.

Reviewer #3 (Remarks to the Author):

The manuscript by Zeng et al. reports the origin and evolution of Qingke barley in Tibet. The

authors generated and collected genomic data for 440 world-wide barley accessions, and have provided clear population genetic evidences (e.g. diversity, low-frequency-allele distribution, LD, D statistics, domesticated gene haplotypes and etc.) to suggest that Tibet Qingke was originated from eastern domesticated barley, and exclude the previous hypothesis that Tibet is an origin center for barley. I think the MS is well written and the results are valuable to the researchers in barley evolutionary biology community. The manuscript could be accepted after some minor revisions.

1. The analyses for origin of Qingke are well performed, and this part could be benefited from further analysis to estimate the effective population size (N_e) for Qingke and its candidate ancestor eastern barley population, and to infer the split time of Qingke from eastern barley. The suggested approaches are MSMC (<https://github.com/stschiff/msmc>.) or SMC++ (<https://github.com/popgenmethods/smcpp/>).

2. I have concerns for the selective sweeps identification. The authors claimed that 'The low diversity region of eastern domesticated barley was likely directly transferred to Qingke and may be considered as a selective sweep of Qingke group.'. So they only remained less than 20% of the genome which is of high diversity in eastern barley population for Tibet qingke adaptive gene identification. I feel this approach may lose lots of true selective loci in Qingke population. According to the authors, the genome-wide genetic diversity of eastern barley is $1.94e-3$ almost double as qingke group ($1.09e-3$). Therefore, there is obvious diversity reduction probably due to a strong genetic bottleneck and further natural or artificial selection. I suggest the authors to perform genome-wide selective sweep analysis using eastern barley as background, and find candidate selective regions for Qingke group.

Point-by-point response to reviewer comments:

Reviewer #1 (Remarks to the Author):

Zeng et al. report a study of nucleotide sequence diversity in Tibetan barley. The primary aim is to determine the route of introduction of barley, a staple crop at high elevations in Tibet. The study involves a truly impressive amount of data. There are various numbers listed in the text, but there are ~180 whole genome sequences from barley that appear to be newly reported here. The authors indicate that there have been a number of prior studies of the origin of domesticated barley in Tibet. One issue has been that apparently wild or feral forms of the progenitor species occur within the region, making the origins of the local cultivated form unclear. The authors seem to largely accomplish their aim of determine the origin of Tibetan barley. However, as noted below, there are some quite fundamental issues that should be addressed.

Comment 1: First, for a paper that is almost entirely population genetic analysis, there are a number of material errors in the analysis. The most fundamental issue is with treatment of sample size. The authors don't report heterozygosity estimates, but a little Google searching suggests barley is typically highly inbred. VCFTools, that the authors used for diversity estimates, only reports Watterson's theta and pairwise diversity for diploid samples, and does not account for inbreeding. Pairwise diversity is not a simple function of sample size; you can't calculate the diploid measure and multiply by two! Sample size issues likely also impact estimates of allele frequency divergence (F_{st}) and principal component analysis. In a similar vein, the ADMIXTURE tool used for genetic assignment makes a Hardy-Weinberg assumption. Be sure to check the specific comments below for the lines in the manuscript where these issues with analysis occur in the text.

Reply: Thank you very much for reminding us to consider the inbreeding of barley. We estimated the heterozygous SNPs proportion in the total SNPs data of each accession and found that most of the samples had < 2% heterozygous SNPs, indicating the highly inbred nature of the studied barley samples. In these re-sequencing papers of soybean (Lam H M et al. 2010; Chung W O N H et al. 2013; Zhou Z et al. 2015), the population genetic analyses were performed considering soybean as common diploid. However, we think your suggestion that "inbred barley samples would seem to be closer to haploid than diploid (Nordborg & Donnelly 1997)" was reasonable for this study. Thus we performed the improvements as follows. (1) The phylogenetic tree and PCA were calculated as the approaches Lam H M et al. (2010, Nature genetics) used for inbred soybean. (2) The individual ancestry coefficients were calculated with sNMF instead of ADMIXTURE. sNMF is reported more appropriate to deal with inbred species and successfully used in barley (Russell J et al. 2016). (3) In population genetics analyses, including nucleotide diversity (π), Watterson's estimator (θ_w), gene diversity/heterozygosity (H_E), recombination rate (ρ), minor allele frequency (MAF) distributions and linkage disequilibrium (r^2), we treated SNPs datasets as haploid, coding all heterozygous sites as missing data. (4) The F_{ST} was calculated using Hudson's estimator with the explicit formula given as equation (10) in Bhatia et al. (2013), which were independent of sample sizes.

Comment 2: Second, there are several errors in the presentation of population genetic results. For example, the authors conflate founder effects and selection when discussing the relative diversity in cultivated barley. Founder effects are demographic events and impact diversity genome-wide (see the Wikipedia article). Selection for adaptation tends to act on individual loci. In a related manner, comparisons on allele frequency divergence are presented as "selective sweep" analysis. While in some scenarios, a selective sweep could result in F_{st} differences, more generally F_{st} differences result from long-term differential selection contributing to local adaptation, a type of selection that is quite distinct from a "selective sweep." Be sure to check the specific comments below for the lines in the manuscript where these issues of presentation occur.

Reply: We agree that we confused the terms of “founder effect” and “selective sweep” in the way we presented the data. We revised this part thoroughly: First, two new paragraphs were written to describe founder effect by (i) genome-wide diversity decrease (ii) demographic history. The founder effect was emphasized in the new version due to the positive demographic analyses by SMC++. Second, the section “Selective event of qingke” was replaced by a paragraph of “candidate genomic region for plateau adaption of qingke”. This section should not be over-emphasized because the exome capture target region covered only 1% of the barley genome, thus SNPs in these regions may not represent or include the major selective gene of the respective selected genomic regions.

Comment 3: Third, the figures could be improved dramatically. For example, the haplotype diagrams convey less information than could be portrayed in a breakdown of haplotypes and their occurrence among accessions.

Reply: We improved the haplotype diagrams by including the gene structure and the global distribution of the haplotype following the example of Russell J et al. (2016).

Comment 4: Fourth, all UNIX shell code used for sequence read mapping, SNP filtering, etc. should be made publicly available in order to increase reproducibility of the work. Github, BitBucket, or a similar online concurrent versioning system is appropriate. I also did not see a statement with regard to the availability of the accessions studied, but hopefully those are also being made availability to barley researchers.

Reply: The custom Perl scripts for data filtering, variants annotation and population genetic analyses were now deposited at sourceforge.net in the “**Code availability**” section. We now described how barley researchers can get accessions to the plant materials in the “**Sample preparation and sequencing**” sections.

Comment 5: Line 56: Founder effect and selection are somewhat different phenomenon.

Reply: The abstract was revised to differentiate properly ‘founder effect’ and ‘selection’.

Comment 6: Line 113: A Google search suggests that genetic evidence of Eastern and Western domesticated barley traces to a Michael Clegg paper. Should that be cited here?

Reply: The accessions used in the Clegg et al. paper were analyzed by Sanger sequencing of 7 nuclear loci totaling 9,296 bp. Line 113 described the accessions used

in this study performed next generation sequencing. So we do not think that the respective reference is a good fit here.

Comment 7: Line 131: Comparisons within species aren't really phylogenetic analyses.

Reply: Yes, the description of “phylogenetic analyses” was not suitable here. We used “population structure” instead of “phylogenetic”.

Comment 8: Line 140: Again, a google search suggests most references to Eastern and Western cultivated barley turn up Michael Clegg papers. I wonder if they should be cited here?

Reply: We agree the paper should be cited here as a reference to the earlier description of differentiation of eastern and western cultivated barley. We have revised the paper. Thanks.

Comment 9: Line 167: There are a lot of different types of “population genetic analysis.” Perhaps be more specific? The following sentences are about comparisons of nucleotide sequence diversity? Is that what the authors are referring to?

Reply: We describe it more specifically now. The new sentence is “Population genetic analysis, including nucleotide diversity (π), Watterson's estimator (θ_w), gene diversity/heterozygosity (H_E), recombination rate (ρ), minor allele frequency (MAF) distributions and linkage disequilibrium (r^2), of the defined barley groups was carried out to test whether Tibet could be recognized as a center of barley diversity and origin as previously suggested³⁻⁷”. The reference of nucleotide diversity (π), Watterson's estimator (θ_w), gene diversity/heterozygosity (H_E) and recombination rate (ρ) were listed in the method.

Comment 10: Line 171: The relative high nucleotide diversity estimates again sent me back to the literature. These numbers seem similar to, though lower than, those from the 2014 Michael Clegg paper from J of Heredity.

Reply: The nucleotide diversity reported by Clegg et al. of 7 loci in 36 cultivated and 45 wild barley was higher than what we present here in our study – most likely this is a bias due to the small number of loci analyzed by Clegg et al. and they didn't declare if their loci were chosen randomly. Common practice in studies based on small numbers of Sanger-sequence-based markers or traditional molecular markers (such as SSR, AFLP, RAPD et al.) was to select loci present in high diversity regions, because high diversity loci were useful to reflect the population differentiation whereas non-polymorphic loci did not allow such analysis.

The nucleotide diversity published for other inbred species is similar with ours. So the result nucleotide diversity in our study is not suspicious.

species	journal	nucleotide diversity (1×10^{-3})	reference
soybean	NG	wild: 2.966; cultivar: 1.894	Lam H M, et al (2010)
sorghum	NC	wild: 3.881; cultivar: 2.514	Mace E S et al (2013)
soybean	NBT	wild: 2.94; landrace: 1.40	Zhou Z et al (2015)
rice	Nature	landrace: ~ 2.0	Wang W et al. (2018)

Comment 11: Line 185: Parametric estimates of recombination rate would be preferable here and would more directly parallel the nucleotide diversity estimates. See LDhat for example.

Reply: The recombination rate ($\rho=4N_e r$) was estimated using a described composite likelihood approach (Hudson R, 2001) with Maxhap on a per-contig level as described previously (Russell J et.al 2016; Hufford M B et al. 2012).

Comment 12: Line 195: There are half a dozen different D statistics in population genetics (Lewontin's D for measuring LD, Tajima's D for site frequency spectrum, etc.). Specify here what is being measured and cite the source of the test, not just the software implementation.

Reply: We now describe the source of the D statistics (D of four-population reported by Durand E Y et al. 2011) in the method section.

Comment 13: Lines 247-253: More discussion around the three possible scenarios brought up here is needed and interpretation of results relating to these three scenarios.

Reply: We re-wrote the paragraph to specifically discuss the possible scenarios.

Comment 14: Lines 274-277: Stronger evidence is needed to claim that the gene HvCRF1 is associated with a selective sweep. Comparing to previously reported genes in other plant species is not strong enough evidence considering all of the genome sizes of the other plants are much smaller than barley. Proportionally, a 14 Mbp stretch is about the same as the other genomes mentioned. Perhaps use SweepFinder (Nielson et al 20015) or XP-CLR (Chen et al 2010) to provide more evidence for this conclusion.

Reply: We used SweepFinder (Nielson et al. 2001) to investigate the selective sweep. The conclusion that HvCRF1 was a major selective genes was not confirmed because many of the F_{ST} -based selective region was overlapped by selective sweep, including HvCRF1 region. We don't know the exact length of the sweep and also don't know which gene was the major selective gene using only exome capture region SNPs. We removed this part and toned it down.

Comment 15: Line 252: The size of the founding population and the strength of selection are likely two separate issues that are being confounded in this presentation. Demography (founder events) tends to impact the whole genome while adaptation tends to cause selection to act on individual loci (Cavalli-Sforza 1966). So in general, adaptation does not impact the whole genome through founder effect.

Reply: Thanks for pointing this out clearly. We rephrased to remove the confusion about “founder effect” and “adaptation”.

Comment 16: Line 320: The statement about the Tibetan population at the beginning of the Methods seems out of place.

Reply: We removed some of the statement about the Tibetan population to the “introduction” section. They are more suitable in the third paragraph of “introduction” section.

Comment 17: Lines 354-357: Include logic on using default hard filtering parameters for SNP and indel sets, especially since this study heavily relies on high confidence SNP calls.

Reply: We described why I used hard filtering approach in the method section. The sentence was “For barley no whole genome SNP and Indel datasets was available to carry on “Base Quality Score Recalibrator” (BQSR) and “Indel Realigner”, so we used the following approach as GATK website recommend for non-human data (<https://gatkforums.broadinstitute.org/gatk/discussion/1706/best-recommendation-for-base-recalibration-on-non-human-data>). First, we did an initial round of variants calling for our original data by command “HaplotypeCaller”. Without available truth/training variants, we used GATK tool “harder filter” to filter the variants instead of Variant Quality Score Recalibration (VQSR) as GATK recommend (<https://software.broadinstitute.org/gatk/documentation/article.php?id=3225>).” The default hard filtering parameters were strict enough for filtering SNPs. In addition, we did Sanger sequencing to estimate SNP quality.

Comment 18: Line 430: It is not clear that there is a biological rationale for calculating diversity only for individuals from a region that show no evidence of admixture. Of course, the diversity in a sample owing to admixture remains an important portion of overall diversity.

Reply: We disagree - the admixed samples had to be removed. For instance, if we want to calculate the diversity of European, the hybrid of European-Chinese should be removed. The hybrid will increase the diversity of European because the Chinese diversity were pulled in.

Comment 19: Line 447: VCFtools calculates thetapi for diploid samples. Inbred

barley samples would seem to be closer to haploid than diploid. See Nordborg & Donnelly 1997 in Genetics. This means the number of chromosomes samples is off by a factor of 2 (in the denominator) in any estimate. Both LD and Fst estimates are also dependent on sample size.

Reply: We made a mistake that the VCF result should be pi not theta-pi. In the new version we calculated pi, as well as LD and Fst considering the SNPs data as diploid. The pi, LD was calculated with the method used in inbred soybean (Lam H M et al. 2010). The F_{ST} was calculated using Hudson's estimator with the explicit formula given as equation (10) in Bhatia et al. (2013), which were independent of sample sizes.

Comment 20: Line 455: The theta-pi estimates are already a parametric estimate. They are adjusted for sample size. Ten sampled chromosomes is generally a sufficient sample size for this type of comparison. See Pluzhnikov and Donnelly 1996 in Genetics.

Reply: We made a mistake that the VCF result should be pi not theta-pi. We do the sample normalization for keeping our study in a strict manner. However, do or don't do resulted the same conclusion.

Comment 21: Line 462: There are 4 or 5 D statistics in population genetics. Cite the original source of the approach along with the software implementation.

Reply: Thanks. We have cited the original source in the method section.

Comment 22: Line 472: Perhaps what the authors want to say here is that “qingke was not directly derived from wild barley”

Reply: Thanks. The paragraph was revised.

Comment 23: Line 475: This type of demographic event is not the same phenomenon as a selective sweep. Selective sweeps are locus-specific phenomena. Check the Wikipedia article at: https://en.m.wikipedia.org/wiki/Selective_sweep. In general, it isn't clear what is going on in this analysis. Why limit to the “diversity in top 20%”? Also, again Fst outliers and selective sweeps may be indicative of quite distinct evolutionary processes.

Reply: The term of “founder effect” and “selection” was described clearly in the new version. The section “Selective event of qingke” was replaced by a paragraph of “candidate genomic region for plateau adaption of qingke”. Limiting “diversity in top 20%” was really not necessary. The restrictions were lifted in the new version.

Reviewer #2 (Remarks to the Author):

The paper makes an interesting and timely contribution in an important area of research. The observations here are largely about clarity, and optimal connectivity between text, figures and tables, in both main and supplementary sections.

Reply: Thanks you very much for your comment to our study.

Comment 1: The tables illustrated in the file 'Supp figs with captions' do not match those in the file 'Qingke tables_162380_0_supp_2868546_p66v17'. Perhaps these tables are all meant to be in supplementary data. Clarification needed

Reply: The Supplementary Data were large Tables as separate sheets in the same Excel file. We renamed each sheet.

Comment 2: The titles/legends for the Supplementary Tables (in file 'Qingke tables_162380_0_supp_2868546_p66v17') are inadequate - there is only a brief one-line description in the top row of the relevant Excel sheet, and sometimes some more data at the bottom of the table. Proper legends are needed with fuller explanations.

Reply: We improved the titles/legends and explanation for each main Figure, main Table, Supplementary Data (large Tables), Supplementary Figure, and Supplementary Tables (small Tables which could be showed in Word file)

Comment 3: The grammar needs improving in the legends. The grammar is OK in the main body of the paper.

Reply: Thanks. We improved it in the new version.

Comment 4: Table S1 – This requires accession status (landrace, cultivar, etc), more details of collection sites, including latitudes and longitudes, where these were available.

Reply: It looks we have provided what you mentioned in Supplementary Data sheet “WGS accessions” and “ES accessions”. Please feel free to let us know if anything else are needed.

Comment 5: The 'Results' section would be more appropriately titled 'Results and Discussion'

Reply: Thanks. We have revised the 'Results' to 'Results and Discussion'.

Comment 6: 'Tibetan weedy barley' as a category lacks clarity - different morphological types are combined within this category – cf. lines 64 to 84, and 110 to

116 of the introduction, and then the section 'Origin of *H. agriocrithon* and Tibetan weedy barleys' in Results. It would be helpful to the reader to separate these types out and call them something different - not necessarily in the analyses, but in the discussion of results.

Reply: We wrote a sentence in the introduction to tell the readers about the derivation of 'Tibetan weedy barley'. The sentence is “It should be noted that while “Tibetan weedy barley” or “Tibetan semi-wild barley” is not a name used in standard barley taxonomy, it has been a popular name used in Tibet by Tibetans or some qingke researchers to distinguish qingke from other Tibetan barleys”.

Reviewer #3 (Remarks to the Author):

The manuscript by Zeng et al. reports the origin and evolution of Qingke barley in Tibet. The authors generated and collected genomic data for 440 world-wide barley accessions, and have provided clear population genetic evidences (e.g. diversity, low-frequency-allele distribution, LD, D statistics, domesticated gene haplotypes and etc.) to suggest that Tibet Qingke was originated from eastern domesticated barley, and exclude the previous hypothesis that Tibet is an origin center for barley. I think the MS is well written and the results are valuable to the researchers in barley evolutionary biology community. The manuscript could be accepted after some minor revisions.

Reply: Thanks you very much for your comment to our study.

Comment 1: The analyses for origin of Qingke are well performed, and this part could be benefited from further analysis to estimate the effective population size (N_e) for Qingke and its candidate ancestor eastern barley population, and to infer the split time of Qingke from eastern barley. The suggested approaches are MSMC (<https://github.com/stschiff/msmc>.) or SMC++ (<https://github.com/popgenmethods/smcpp>).

Reply: Thank you very much for reminding us to perform demographic history analyses. The MSMC relies on deep whole genome sequencing data. The average $\sim 10\times$ coverage of our accessions don't support MSMC analyses. SMC++ is based on whole genome SNPs and a recent report had shown its good performance in *Brachypodium distachyon* population data (Stritt C, 2017). So we tried to use SMC++ and found very good results. The detailed analysis is presented in the new version.

Comment 2: I have concerns for the selective sweeps identification. The authors claimed that ‘The low diversity region of eastern domesticated barley was likely directly transferred to Qingke and may be considered as a selective sweep of Qingke group.’. So they only remained less than 20% of the genome which is of high diversity in eastern barley population for Tibet qingke adaptive gene identification. I

feel this approach may lose lots of true selective loci in Qingke population. According to the authors, the genome-wide genetic diversity of eastern barley is $1.94e-3$ almost double as qingke group ($1.09e-3$). Therefore, there is obvious diversity reduction probably due to a strong genetic bottleneck and further natural or artificial selection. I suggest the authors to perform genome-wide selective sweep analysis using eastern barley as background, and find candidate selective regions for Qingke group.

Reply: We have only 7 WGS eastern landrace accessions. Furthermore the 7 accessions were only distributed in eastern China and can't represent the diversity of eastern barley, especially the eastern barley in Southern Asia which evolved into qingke. The accessions limitation made it difficult to accurately reflect the selective region using only 7 WGS eastern barley as background. Thus we also used the eastern barley including both WGS and ES samples as background. The section "Selective event of qingke" was replaced by a paragraph of "candidate genomic region for plateau adaption of qingke", and the local adaption of qingke was shown by F_{ST} and CLR as recommended by reviewer1.

Reviewers' comments:

Reviewer #1 (Remarks to the Author):

- Major points in revision:

- 1) I did not mention it earlier, but the switch to BWA MEM is an upgrade in read mapping.
- 2) Again, not mentioned in the previous review, but probabilistic approaches to estimation of sequence descriptive statistics are generally preferable to SNP calls. Filtering tends to be arbitrary. See ANGSD for an improved approach (Korneliussen T, Albrechtsen A, Nielsen R. 2014. ANGSD: Analysis of Next Generation Sequencing Data. BMC Bioinformatics. 15: 356.)
- 3) Typically shared polymorphism with an outgroup is the result of incomplete coalescence (also known as lineage sorting) rather than gene flow. For a good estimation of the time to monophyly, see: Rosenberg NA. 2003. The shapes of neutral gene genealogies in two species: probabilities of monophyly, paraphyly, and polyphyly in a coalescent model. *Evolution*. 57: 1465-1477.
- 4) Removing the structural variation analysis makes for a more coherent story.

Reviewer #1

- Comment 4: The sourceforge.net page is helpful, but it primarily includes the Perl scripts used for calculation of nucleotide sequence diversity. To reproduce the work reported, the shell scripts used for read mapping and SNP quality filtering, for example, would also be needed.

- Comment 10: Diversity within a species is determined by the effective population size, which is a function of the census size. The mating system has a relatively small effect on nucleotide diversity. Mating system can reduce estimates of nucleotide diversity by a factor of ~2 (there are several related papers by Magnus Nordborg and colleagues, see Nordborg 2000 *Genetics*). So you can't readily infer nucleotide diversity in one species by comparison to another species with a similar mating system.

So, more Google searching uncovers a new book edited by one of the coauthors here. In chapter 17 "The Barley Genome", Table 17.1, has a summary of barley diversity. Based on the studies from Sanger sequencing, cultivated barley average $\theta = 0.005$, whereas the values reported from one prior study using exome capture are 0.00153. Unfortunately, the Russell et al. 2016 paper mentioned several times here does not report θ values. So, is the difference due to biased selection of loci in Sanger studies or some aspect of Illumina resequencing? It appears that prior Illumina resequencing studies in plants have also reported lower values of θ than in Sanger-based. A notable example is the Hufford et al. 2012 *Nature Genetics* study of maize. One potential issue relates to Major point 1) above. The mismatch parameter for BWA MEM is tuned for human data, with about 1 mismatch / 1000 bp or $\theta = 0.001$. Not increasing this value will under-map reads at more diverse loci and result in a tendency toward lower estimates of θ . Also, for loci with large amounts of missing data, the missing data may be non-random, meaning that more divergent haplotypes are missing. This results in lower estimates of diversity. As noted in Comment 4, a complete archive of the scripts needed to reproduce the work would be helpful, including those used for read mapping with BWA-MEM and for variant filtering. Finally, I would note that reporting summaries of nucleotide sequence diversity measures is needed. There are many pages of figures with graphs of these statistics, but no summary table.

- Comment 17: Given that the authors note a lack of published variants for barley, they should provide their community with VCF files resulting from their SNP calling process. This will help improve variant quality score recalibration in future studies. Often VCF files are too large for a Github or similar repository, but should be made available on Figshare or similar sites.

- Comment 18: With regard to admixture and diversity, if historical admixture has contributed to diversity in a population, that should be quantified. If one measures diversity in a human population, in Toronto, Canada or São Paulo, Brazil, the people have ancestry from around the world. The nucleotide diversity that is measured results in part from admixture and there is no

rationale for removing admixed individuals.

- Comment 19: The response here is really confusing. Elsewhere, including in the Perl code, the authors say that diversity estimates are now for haploid sampling. But here, it says the values are from SNP data as DIPLOID? Also, it is not at all clear what is meant by "We made a mistake that the VCF result should be pi not theta-pi." Are the authors talking about a per locus measure? The sentence, "The pi, LD was calculated with the method used in inbred soybean..." doesn't really provide clarity. Was a diploid or haploid sample number used?

- Comment 20: Again, I don't understand this response at all. "We made a mistake that the VCF result should be pi not theta-pi." As stated earlier, the estimates of population parameters like pi and Watterson's theta are already adjusted for sample size.

Minor comments:

- Line 76 - Do the authors mean "edges" of fields rather than "ridges?"
- Line 136 - "Indel" is not a proper noun and should not be capitalized.
- Line 176 - Absolute numbers for these descriptive statistics should be reported, as they can be compared to studies in barley and other species.
- Line 186 - Again, reporting the average values for rho is important, as it makes it possible to compare values to other studies and to other barley populations.
- Line 224 - Again, numbers are useful. Average pi at silent sites, for example, is an excellent comparator among populations. "Lower" or "higher" is obviously entirely relative.
- Line 238 - Again, why say the "effective population size didn't decrease" when numbers can be used to express $\theta = 4N_e\mu$ from the data?

Reviewer #2 (Remarks to the Author):

This paper makes an important contribution to our understanding of prehistoric crop movement in prehistory. As the result of a range of recent archaeobotanical and genetic work, that movement is now understood in broad outline, but resolving the deep history of Qingke makes a significant contribution to that story.

The analyses seem robust and clearly presented, the conclusions well argued and persuasive (and important in the context of different published about the geography of barley domestication). The methodology seems appropriate, and the various analytical procedures valid and clearly documented.

Reviewer #3 (Remarks to the Author):

The authors have revised the manuscript according to our suggestions, I agree to accept the revised manuscript.

Rebuttal to reviewer comments

Reviewers' comments:

Reviewer #1 (Remarks to the Author):

- Major points in revision:

- 1) I did not mention it earlier, but the switch to BWA MEM is an upgrade in read mapping.
- 2) Again, not mentioned in the previous review, but probabilistic approaches to estimation of sequence descriptive statistics are generally preferable to SNP calls. Filtering tends to be arbitrary. See ANGSD for an improved approach (Korneliussen T, Albrechtsen A, Nielsen R. 2014. ANGSD: Analysis of Next Generation Sequencing Data. BMC Bioinformatics. 15:356.)

Response: We performed whole-genome shotgun sequencing to 10-fold coverage and integrated with published exome capture data which provided ≥ 10 -fold coverage in target regions. ANGSD was designed specifically with low-coverage resequencing (1x) in mind. GATK, which is a commonly used tool applied in many resequencing studies (e.g. the rice 3000 genomes project), achieves high accuracy if sequencing depth is high enough (see recent benchmark by Li et al. 2018, Nature Methods). Also note that not all downstream tools can work with probabilistic genotype calls of ANGSD.

- 3) Typically shared polymorphism with an outgroup is the result of incomplete coalescence (also known as lineage sorting) rather than gene flow. For a good estimation of the time to monophyly, see: Rosenberg NA. 2003. The shapes of neutral gene genealogies in two species: probabilities of monophyly, paraphyly, and polyphyly in a coalescent model. Evolution. 57:1465-1477.

Response: Shared polymorphisms were the reason why we did not use *H. bulbosum* as an outgroup. According to Brassac and Blattner, 2015, Systematic Biology, *H. bulbosum* and *H. vulgare* diverged less than 5 million years ago. Joint analysis of a barley diversity panel and four *H. bulbosum* samples (unpublished results, M. Mascher) indicated that ~ 10 % of sites polymorphic in *H. vulgare* are shared with *H. bulbosum*. *H. vulgare* and *H. bulbosum* are sympatric and gene-flow between them may have taken place, which could also explain the occurrence of shared polymorphism. By contrast, *H. pubiflorum* is among the *Hordeum* species most distantly related to *H. vulgare*. Occurring only in South America, it diverged from the *H. vulgare* lineage about 9 million years ago (Brassac and Blattner, *ibid.*).

- 4) Removing the structural variation analysis makes for a more coherent story.

Response: we agree.

Reviewer #1

- Comment 4: The sourceforge.net page is helpful, but it primarily includes the Perl scripts used for calculation of nucleotide sequence diversity. To reproduce the work reported, the shell scripts used for read mapping and SNP quality filtering, for example, would also be needed.

Response: We have uploaded the in-house Perl and Shell scripts used in this study to the web (<https://sourceforge.net/projects/origin-of-qingke-barley/files/?source=navbar>), including six

compressed files: 01.reads_mapping, 02.mapping_statistics, 03.GATK_variant_calling, 04.variant_quality_filtering, 05.variant_annotation and 06.population_genetics_statistics.

- Comment 10: Diversity within a species is determined by the effective population size, which is a function of the census size. The mating system has a relatively small effect on nucleotide diversity. Mating system can reduce estimates of nucleotide diversity by a factor of ~ 2 (there are several related papers by Magnus Nordborg and colleagues, see Nordborg 2000 Genetics). So you can't readily infer nucleotide diversity in one species by comparison to another species with a similar mating system. So, more Google searching uncovers a new book edited by one of the coauthors here. In chapter 17 "The Barley Genome", Table 17.1, has a summary of barley diversity. Based on the studies from Sanger sequencing, cultivated barley average $\theta = 0.005$, whereas the values reported from one prior study using exome capture are 0.00153. Unfortunately, the Russell et al. 2016 paper mentioned several times here does not report θ values. So, is the difference due to biased selection of loci in Sanger studies or some aspect of Illumina resequencing? It appears that prior Illumina resequencing studies in plants have also reported lower values of θ than in Sanger-based. A notable example is the Hufford et al. 2012 Nature Genetics study of maize.

One potential issue relates to Major point 1) above. The mismatch parameter for BWA MEM is tuned for human data, with about 1 mismatch / 1000 bp or $\theta = 0.001$. Not increasing this value will under-map reads at more diverse loci and result in a tendency toward lower estimates of θ . Also, for loci with large amounts of missing data, the missing data may be non-random, meaning that more divergent haplotypes are missing. This results in lower estimates of diversity.

Response: We have added a cautionary note at the end of the Results/Discussions section to point at potential shortcomings of analyzing sequence variation using short read mapping. We performed a brief literature survey on the use of BWA-MEM. Notably, we found that Maize Hapmap3 project (Bukowski, 2017, Gigascience) used BWA-MEM to align. Presumably, also two recent studies in *Drosophila* species (*D. serrata*, Reddix et al., 2018, G3; *D. simulans*, Signor et al. GBE) used BWA-MEM with default parameters as no specific parameter or steps for parameter optimization are described.

We believe that a direct comparison between diversity estimates from Sanger and Illumina sequence data may be difficult because the Sanger sequencing can interrogate only a small portion of loci that may not be sampled randomly. In particular, about one-third of barley genes reside in the low-recombining, low-diversity pericentromeric regions which at one point were thought to be devoid of genes and maybe thus underrepresented in Sanger datasets. Our own analyses and the results of Russell et al. 2016 indicate that diversity between distal and proximal regions of the genome can vary by one order of magnitude. So, simple comparison of averaged \$\theta\$ values may not be a worthwhile exercise.

As noted in Comment 4, a complete archive of the scripts needed to reproduce the work would be helpful, including those used for read mapping with BWA-MEM and for variant filtering.

Response: Please look at the response for Comment 4.

Finally, I would note that reporting summaries of nucleotide sequence diversity measures is needed. There are many pages of figures with graphs of these statistics, but no summary table.

Response: Please look at the content of Table 1 of manuscript. It reported genome-wide and exome-wide per bp value of different nucleotide sequence measures in each barley group. Please also look at Supplementary Table 7, 8, 9 and 10, they reported genome-wide and exome-wide per bp value of diversity on chromosomal level. We think these tables had reported the summaries of nucleotide sequence diversity.

- Comment 17: Given that the authors note a lack of published variants for barley, they should provide their community with VCF files resulting from their SNP calling process. This will help improve variant quality score recalibration in future studies. Often VCF files are too large for a Github or similar repository, but should be made available on Figshare or similar sites.

Response: We have registered a Digital Object Identifier (DOI) with the Plant Genomics and Phenomics Research Data Repository (PGP, <http://edal-pgp.ipk-gatersleben.de>). A temporary URL (<https://doi.ipk-gatersleben.de/DOI/31afe8ad-8421-4354-acd3-7281cfd63abd/8e2b74e5-3297-487d-8752-f57441f0bc5a/2/1847940088>) is provided in the manuscript. The final DOI (resolvable with dx.doi.org) will be activated upon acceptance of the manuscript.

- Comment 18: With regard to admixture and diversity, if historical admixture has contributed to diversity in a population, that should be quantified. If one measures diversity in a human population, in Toronto, Canada or São Paulo, Brazil, the people have ancestry from around the world. The nucleotide diversity that is measured results in part from admixture and there is no rationale for removing admixed individuals.

Response: We added information about the rationale for excluding admixed samples in the Methods section.

- Comment 19: The response here is really confusing. Elsewhere, including in the Perl code, the authors say that diversity estimates are now for haploid sampling. But here, it says the values are from SNP data as DIPLOID? Also, it is not at all clear what is meant by "We made a mistake that the VCF result should be pi not theta-pi." Are the authors talking about a per locus measure? The sentence, "The pi, LD was calculated with the method used in inbred soybean..." doesn't really provide clarity. Was a diploid or haploid sample number used?

Response: In the initial version of the manuscript VCFtools were used with command "--site-pi" to calculate the nucleotide diversity (pi). The description about this command "--site-pi" in VCFtools manual is (http://vcftools.sourceforge.net/man_latest.html): --site-pi, measures nucleotide divergence on a per-site basis. The output file has the suffix ".sites.pi". Thus the obtained value should have been called

nucleotide divergence (π) rather than θ - π . By mistake we considered it as θ - π . Also our response sentence "In the new version we calculated π , as well as LD and F_{st} considering the SNPs data as diploid" was wrong. It should read "In the new version we calculated π , as well as LD and F_{st} considering the SNPs data as haploid" (see the method section and our Perl script). The sentence "The π , LD was calculated with the method used in inbred soybean (Lam H M et al. 2010)" was improper here because the paper (Lam H M et al. 2010) calculated π and LD as diploid. We removed this sentence and apologize for the mistake.

- Comment 20: Again, I don't understand this response at all. "We made a mistake that the VCF result should be π not θ - π ." As stated earlier, the estimates of population parameters like π and Watterson's θ are already adjusted for sample size.

Response: We apologize for causing this confusion. Please see our response to Comment 19. In addition, we have removed the analyses of sample size adjusting, as well as supplementary Figure 12 when comparing diversity of Tibetan weedy barley with other barley groups. As your suggestion, it is not necessary to do samples normalization and to plot the quartile statistics box. So we directly considered 10 Tibetan weedy barley samples as a group, calculating exome-wide π and θ_w . We compared the per bp value of π and θ_w in Tibetan weedy barley with other barley groups. We revised the sentence about the comparing descriptions as "Ten Tibetan weedy barleys showed slightly higher genetic diversity ($\pi=1.43e-3$, $\theta_w=1.37e-3$) than qingke (Table 1), but lower than other domesticated barleys (Table 1), and also showed domestication gene haplotypes that were shared with domesticated barley."

Minor comments:

- Line 76 - Do the authors mean "edges" of fields rather than "ridges?"

Response: Yes, what we meant "edges". We have replaced ridges by edges.

- Line 136 - "Indel" is not a proper noun and should not be capitalized.

Response: we used now "INDELS" throughout the manuscript.

- Line 176 - Absolute numbers for these descriptive statistics should be reported, as they can be compared to studies in barley and other species.

Response: Please look at the response for Comment 10. These numbers are reported in Table 1. The per bp value in barley can be compared to other species.

- Line 186 - Again, reporting the average values for ρ is important, as it makes it possible to compare values to other studies and to other barley populations.

Response: Please look at the response for Comment 10. These numbers are reported in Table 1.

- Line 224 - Again, numbers are useful. Average pi at silent sites, for example, is an excellent comparator among populations. "Lower" or "higher" is obviously entirely relative.

Response: Please also look at the response for Comment 10. These numbers are reported in Table 1 and Supplementary Table 7, 8, 9, 10.

- Line 238 - Again, why say the "effective population size didn't decrease" when numbers can be used to express $\theta = 4N_e\mu$ from the data?

Response: We agree. We have used the value of $\theta = 4N_e\mu$ instead of the simple expression "effective population size didn't decrease". The sentence in the manuscript has been altered as "Over the same period (~2,000 to ~4,500 cal yr B.P.), the effective population size value of the eastern Asian landraces, which entered central China around Tibet (route I, Fig. 1c), remained constant between ~4,374 to ~4,500 (Fig. 4b; Supplementary Data), indicating the Tibetan Plateau environment provided the main factor resulting in a founder effect.". We added the underlying data for Fig. 4b in the Supplementary Data to show the precise effective population size during the period.

Reviewer #2 (Remarks to the Author):

This paper makes an important contribution to our understanding of prehistoric crop movement in prehistory. As the result of a range of recent archaeobotanical and genetic work, that movement is now understood in broad outline, but resolving the deep history of Qingke makes a significant contribution to that story.

The analyses seem robust and clearly presented, the conclusions well argued and persuasive (and important in the context of different published about the geography of barley domestication).

The methodology seems appropriate, and the various analytical procedures valid and clearly documented.

Response: We thank the Reviewer for his/her efforts.

Reviewer #3 (Remarks to the Author):

The authors have revised the manuscript according to our suggestions, I agree to accept the revised manuscript.

Response: We thank the Reviewer for his/her efforts.

Reviewers' comments:

Reviewer #1 (Remarks to the Author):

- First, I apologize for asking the authors for elements of the manuscript that were included! Unfortunately, Table 1, is not in the PDF of the manuscript, so it was difficult to find. So, I'm seeing Table 1 for the first time. In Table 1, π and Watterson's θ are both estimates of $4N_e$. So they should be somewhat similar. In the case of most of the samples, that is true, but for the wild samples they are very different. Why is that? Tajima's D examines this relationship. It is a very commonly reported sequence summary. What is the average value of Tajima's D for these samples? If you look at the site frequency spectrum, is it clear why the Watterson's θ value is so much higher for wild samples than the estimate of π ?

- A more appropriate comparison of ANGSD and SNP calls is available from Han et al. 14 Molecular Biology and Evolution. Of course, coverage can be low for individual genomic segments even in high coverage data. The SNP call-based analysis here is appropriate, but as approaches evolve, the authors are encouraged to explore updates like probabilistic approaches to diversity estimates.

- Also related to Table 1, ρ and θ values are on the same scale, both estimates of $4N_e$. It can be useful to compare ρ/θ as a means of understanding demographic history. See for example Haddrill et al. 2005 Genome Research. It looks like Michael Clegg reported similar results for barley in a 2006 Genetics paper. Are these results similar in magnitude?

- The availability of all scripts used for processing is a nice upgrade. Consider in the future using a concurrent versioning system repository like Github or BitBucket.

- The authors have clarified the issues regarding diploid versus haploid sampling.

- I don't understand some of the language in the response regarding π and $\theta-\pi$, but the analysis appears correct.

- With regard to descriptive statistics, Tajima's D is generally a more useful statistic than the gene diversity/heterozygosity reported in Table 1 and Supplementary Table 9? What do we learn from this estimate? The values seem extremely similar in magnitude and direction to estimates of π .

- The manuscript is improved over the previous version. The authors have clarified the most confusing components, like mentioning both haploid and diploid estimates of descriptive statistics.

Reviewers' comments:

Reviewer #1 (Remarks to the Author):

Comment #1: First, I apologize for asking the authors for elements of the manuscript that were included! Unfortunately, Table 1, is not in the PDF of the manuscript, so it was difficult to find. So, I'm seeing Table 1 for the first time. In Table 1, π and Watterson's theta are both estimates of $4N_e\mu$. So they should be somewhat similar. In the case of most of the samples, that is true, but for the wild samples they are very different. Why is that? Tajima's D examines this relationship. It is a very commonly reported sequence summary. What is the average value of Tajima's D for these samples? If you look at the site frequency spectrum, is it clear why the Watterson's theta value is so much higher for wild samples than the estimate of π ?

Response:

We apologize for making access to Table 1 not obvious to the reviewers and appreciate the additional instructions. The wild barley group has more segregating sites than the domesticated groups, and most of these are low-frequency variants (see Supplementary Fig 8 in the new submitted version). As expected, we have a negative Tajima's D in the wild group due to the excess of rare variants. We have added the estimates for all groups to Table 1 and added the note in the sentence of manuscript "The wild group had the highest proportion of low-frequency-alleles (also supported by the negative Tajima's D, Table 1), while the qingke group had the lowest (Supplementary Fig. 8)". The Perl script for calculating Tajima's D has been uploaded.

Revised Table 1:

diversity	overlapped SNPs data				WGS SNPs data	
	wild	western	eastern	qingke	western cultivars	qingke
π	3.03	2.06	1.75	1.10	2.77	1.49
θ_w	5.39	2.07	1.57	1.00	2.79	1.47
H_E	2.85	1.94	1.52	0.99	2.51	1.36
ρ	2.85	0.23	0.07	0.04	0.32	0.06
Tajima's D	-1.16	0.40	1.23	0.86	0.39	0.53

Comment '2: A more appropriate comparison of ANGSD and SNP calls is available from Han et al. 14 Molecular Biology and Evolution. Of course, coverage can be low for individual genomic segments even in high coverage data. The SNP call-based analysis here is appropriate, but as approaches evolve, the authors are encouraged to explore updates like probabilistic approaches to diversity estimates.

Response: We appreciate the reviewer's comment "The SNP call-based analysis here is appropriate..." and thus disagree that further explorations into other tools (which are not explicitly specified) is a justified request at this stage as there is no indication that additional such analysis will lead us to other conclusions. We agree that as soon as our data is published, it may

be attractive and worthwhile to other researchers in the community to test newly emerging tools on this dataset.

Comment #3: Also related to Table 1, rho and theta values are on the same scale, both estimates of $4N_e$. It can be useful to compare rho/theta as a means of understanding demographic history. See for example Haddrill et al. 2005 Genome Research. It looks like Michael Clegg reported similar results for barley in a 2006 Genetics paper. Are these results similar in magnitude?

Response: Similar to our results, the rho/theta in wild barley is smaller than 1 for most of the 17 loci reported in Table 5 of Morrell et al. 2006 Genetics. To investigate demographic history, we used SMC++, which also derive estimates for past N_e taking nucleotide diversity and recombination rate into account.

Comment #4: The availability of all scripts used for processing is a nice upgrade. Consider in the future using a concurrent versioning system repository like Github or BitBucket.

Response: We thank the reviewer for this comment and will consider his/her recommendation in the future.

Comment #5: The authors have clarified the issues regarding diploid versus haploid sampling.

Response: We thank the reviewer for acknowledging this.

Comment #6: I don't understand some of the language in the response regarding pi and theta-pi, but the analysis appears correct.

Response: We thank the reviewer for acknowledging this.

Comment #7: With regard to descriptive statistics, Tajima's D is generally a more useful statistic than the gene diversity/heterozygosity reported in Table 1 and Supplementary Table 9? What do we learn from this estimate? The values seem extremely similar in magnitude and direction to estimates of pi.

Response: See the definition for gene diversity (Nei M 1973): "gene diversity/heterozygosity is a measure of genetic variation of a population". Based on its formula (<http://www.uwyo.edu/dbmcd/molmark/lect04/lect4.html>, Eqn 4.1), it accounts for both richness and evenness of polymorphic sites, thus it is a measure to estimate genetic diversity of a population. The formula for pi and gene diversity are different. We reported gene diversity to add a measure of estimating genetic diversity for barley groups. Considering the figure for pi and gene diversity based on 10Kb value are very similar (Supplementary Fig 4 and 6), we carefully checked the data of them and found we didn't make any mistake. The values of them in each window are different but showed very similar trend across the genome plotted by Gnuplot. We have uploaded the underlying table of pi and gene diversity in 1H chromosome and the Shell script to show how we plotted the figure (06.population_genetics_statistics/pi_He_plot_test in

<https://sourceforge.net/projects/origin-of-qingke-barley/files/?source=navbar>.
For Tajima's D, see our response above.

Comment #8: The manuscript is improved over the previous version. The authors have clarified the most confusing components, like mentioning both haploid and diploid estimates of descriptive statistics.

Response: We thank the reviewer for acknowledging this.

REVIEWERS' COMMENTS:

Reviewer #1 (Remarks to the Author):

The updates to the manuscript are appropriate.